

# Multicopter measurements of volcanic gas emissions at Masaya (Nicaragua), Turrialba (Costa Rica) and Stromboli (Italy) volcanoes: Applications for volcano monitoring and insights into halogen speciation

Julian Rüdiger[1,*], Lukas Tirpitz[2], J. Maarten de Moor[3], Nicole Bobrowski[2,4,5], Alexandra Gutmann[1], Marco Liuzzo[6], Martha Ibarra[7] and Thorsten Hoffmann[1]

[1]Johannes Gutenberg-University, Institute of Inorganic and Analytical Chemistry, Mainz, Germany
[2]Institute for Environmental Physics, Heidelberg
[3]Observatorio Vulcanológico y Sismológico de Costa Rica, Heredia, Costa Rica
[4]Johannes Gutenberg-University, Institute of Geosciences, Mainz, Germany
[5]Max Planck Institute for Chemistry, Mainz, Germany
[6]Istituto Nazionale di Geofisica e Vulcanologia, Sezione di Palermo, Italy
[7]Instituto Nicaragüense de Estudios Territoriales, Managua, Nicaragua

* now at University of Bayreuth, Atmospheric Chemistry, Bayreuth, Germany.

*Correspondence to*: Julian Rüdiger (j.ruediger@uni-mainz.de)

**Abstract.** Volcanoes are a natural source of several reactive gases (e.g. sulfur and halogen containing species), as well as non-reactive gases (e.g. carbon dioxide). Besides that, halogen chemistry in volcanic plumes might have important impacts on atmospheric chemistry, carbon to sulfur ratios and sulfur dioxide fluxes are important established parameters to gain information on subsurface processes. In this study we demonstrate the successful deployment of a multirotor UAV (quadcopter) system with custom-made lightweight payloads on board for the compositional analysis and gas flux estimation of volcanic plumes. The various applications and their potential with such new measurement strategy are presented and discussed on example studies at three volcanoes encompassing flight heights of 450 m to 3300 m and various states of volcanic activity. Field applications were performed at Stromboli Volcano (Italy), Turrialba Volcano (Costa Rica) and Masaya Volcano (Nicaragua). Two in-situ gas-measuring systems adapted for autonomous airborne measurements, based on electrochemical and optical detection principles, as well as an airborne sampling unit, are introduced. We show volcanic gas composition results including, abundances of $CO_2$, $SO_2$ and halogen species. The new instrumental set-ups were compared with established instruments during ground-based measurements. For total $SO_2$ flux estimations a small differential optical absorption spectroscopy (DOAS) system measured $SO_2$ column amounts on transversal flights below the plume, showing the potential to replace ground-based manned operations.

At Stromboli volcano, short-term fluctuation of the $CO_2/SO_2$ ratios could be determined and confirm an increased $CO_2/SO_2$ ratio in spatial and temporal proximity to explosions by airborne in-situ measurements. Reactive bromine to sulfur ratios of 0.19 x $10^{-4}$ to 9.8 x $10^{-4}$ were measured in-situ in the plume of Stromboli volcano downwind of the vent.



## 1 Introduction

Gaseous volcanic emissions consist of a variety of different compounds and are dominated by water vapor ($H_2O$), carbon dioxide ($CO_2$), sulfur dioxide ($SO_2$), and hydrogen sulfide ($H_2S$) (Symonds et al., 1994). Minor abundant, but nonetheless important gas species are halogen-bearing compounds which are emitted as hydrogen halides (HF, HCl, HBr and HI) and later

partly transformed by heterogeneous reactions into other halogen species, such as bromine monoxide (BrO) or chlorine dioxide (OClO) (Bobrowski et al., 2007). The relative gas composition varies with the types of volcanoes and magmas as well as with transport and degassing mechanisms. Changes in the magma degassing behavior and/or the hydrothermal systems beneath volcanoes generally influence the gas composition and gas fluxes. Measuring the emitted gas composition can provide crucial information on understanding subsurface processes related to activity changes (e.g. Allard et al., 1991; Aiuppa et al., 2007;

Bobrowski and Giuffrida, 2012; de Moor et al., 2016a; Liotta et al., 2017) and help to estimate fluxes of the geological carbon cycle (e.g. Burton et al., 2013) and tectonic processes controlling volcanic degassing (e.g. Aiuppa et al., 2017). In the field of volcanic monitoring, the observation of gas composition changes became an important tool for detecting precursory processes for volcanic eruptions. It has been shown that enhanced $CO_2/SO_2$ emission ratios appear prior to eruptive volcanic activity within the timescale of hours to weeks (e.g. Giggenbach, 1975; Aiuppa et al., 2007; de Moor et al., 2016b). In-situ

measurements of this gas ratio has become a well-established method using electrochemical ($SO_2$) and infrared ($CO_2$) sensors, implemented in so-called Multi-GAS (MG) instruments, which may also contain other sensors and are field-deployable to work autonomously close to volcanic emission sources (Shinohara, 2005; Aiuppa et al., 2006).

Another important parameter for the characterization of volcanic activities are emission rates (fluxes). Particularly, the determination of $SO_2$ fluxes has become a standard procedure by traversing the plume and multiplying the integrated $SO_2$

cross-section with the estimated plume transport speed (e.g. McGonigle et al., 2002; Galle et al., 2003; López et al., 2013). With the development of small DOAS instruments, traversing the plume is not only feasible by cars, boats, or manned aircrafts, but also by walking in case of poorly accessible terrains.

Furthermore, knowledge about in-plume chemical reactions can be drawn from compositional assessment of the gases, which also helps understanding their impact on atmospheric chemistry (e.g. Lee et al., 2005; von Glasow, 2010; Gliß et al., 2015).

Especially the halogen chemistry is of great interest as $BrO/SO_2$ ratios in volcanic plumes are readily measurable by remote sensing UV spectrometry and have been discussed in recent years as another potential precursory observable for volcanic activity changes. It was observed that the $BrO/SO_2$ ratio decreases in advance to eruptive phases (Lübcke et al., 2014) and is lower during periods of continuous activity (Bobrowski and Giuffrida, 2012). However, BrO is not a directly emitted species rather than the product of complex heterogeneous chemistry in the volcanic plume involving reactions with magmatic gases

with entrained air (e.g. Gerlach, 2004, Bobrowski et al., 2007). The variation of BrO by plume age and a transversal distribution in the plume for this species was observed by differential optical absorption spectroscopy (DOAS) measurements (Bobrowski and Platt, 2007). Furthermore, other reactive halogen species with oxidation states $\neq$ -1 (e.g. $Br_2$, $Cl_2$, BrCl and others) have been measured in-situ in the plume of Mt. Etna, Italy (Rüdiger et al., 2017) and Mt Nyamuragira (Bobrowski et al. 2017,





accepted). In the last decade, several model studies (e.g. Bobrowski et al., 2007, Roberts et al., 2009; von Glasow, 2010; Roberts et al., 2014; Jourdain et al., 2016) have engaged on the variation of halogen variability in volcanic plumes with respect to various atmospheric and magmatic parameters. In the case of bromine, it was modelled that the initial emitted hydrogen bromide is depleted shortly after emission under consumption of tropospheric ozone and is transformed to reactive species

such as BrO, HOBr, $Br_2$, BrCl and $BrONO_2$. Due to the challenging task of accessing volcanic plumes on a timescale of minutes after emission, and the lack of spectroscopic methods for most of these reactive species, uncertainties about their relative abundances still exist. One approach towards the in-situ observation of reactive halogen species is the application of gas diffusion denuder sampling using a selectively reactive organic coating (1,3,5-trimethoxybenzen, TMB) to trap and enrich gaseous species containing a halogen atom with the oxidations state +1 or 0 (e.g. $Br_2$ or BrCl), while being insensitive to the

particle phase (Rüdiger et al., 2017).

In the case of most volcanoes sampling on the crater rim presents great logistical challenges and hazards for people and instruments. During phases of high activity crater rims are usually not accessible at all and even during quiescent degassing work at the crater rim represents a considerable risk. However, the knowledge of plume gas composition is an important component for activity assessments of volcanoes (e.g. Carroll and Holloway, 1994; Aiuppa et al., 2006) and therefore gas

monitoring stations are deployed and maintained at the crater rim by researchers, putting themselves at risks. Advancements in the application of remote sensing techniques have helped to minimize personnel exposition to the volcanic danger zone (e.g. Galle et al., 2003; Tamburello et al., 2011). However, still today the detection of certain gas species and/or total amounts of all species of an element is not possible remotely (neither ground based nor with satellites) and therefore in-situ measurements are an important tool, especially for resolving chemical reactions and speciation changes in aging volcanic plumes. With an

in-situ sampling strategy, obtaining samples from the freshly emitted plume is feasible, while ground-based in-situ sampling of the aged plume further downwind is rarely possible and dependent on specific wind and geographical conditions. In the last decade with the development of compact and cost effective unmanned aerial vehicles (UAV) several deployments of gas sensors and other in-situ methods (e.g. particle detection (Altstädter et al., 2015)) as well as applications of spectrometers were realized (e.g. McGonigle et al., 2008, Diaz et al., 2015, Mori et al., 2016, Villa et al., 2016 and references therein). While these

applications focused mostly on the use of sensors and spectroscopy methods, sampling (e.g. canister sampling (Chang et al., 2016)) of the plume has not been reported for volcanoes. Here we present a low-cost UAV-deployable sampling (gas diffusion denuder) and sensing (electrochemical/optical sensors) systems for the determination of $CO_2$, $SO_2$ and halogen species. Our system enabled us to access the plume close to an active vent as well as the aged plume several km downwind of the source and elevated from ground, without exposing operators to the risks in proximity to active vents or employing manned aircrafts

to potential engine-damaging ash and gas plumes. In addition to that, the UAV-deployment of a lightweight DOAS instrument for $SO_2$ flux estimations is presented herein, which enables fast plume traversing in terrains that are usually not accessible by cars or even by foot.



## 2. Site description

### 2.1 Stromboli

Stromboli volcano, the northernmost island of the Aeolian volcanic arc (Italy), rises 924 m above sea level (a.s.l.). The island represents the top part of a large 2500-m-high stratovolcano emerging from the Tyrrhenian Sea floor. Stromboli is well known

5 for its regular (~ every 10-20 min) explosive activity (Strombolian activity). Intermittently, continuous passive degassing occurs from the active vents, which are located in the so-called crater terrace at about 750 m a.s.l.. Ejected lava material is dominantly deposited in a northwestern direction, forming a hardly safe to non-accessible horseshoe-shaped area (Sciara del Fuoco). The summit above the craters is well accessible and characterized by a numerous amount of monitoring stations continuously observing the activity. Thus, Stromboli has been a laboratory volcano for studying magma degassing processes

10 (Allard et al. 2008) and field-testing new instrumentations for many years.

While most gas monitoring station positions benefit from a north westerly wind direction, a south easterly wind only allows gas plume detection by spectroscopic methods. In this study, due to the dominance of southeast winds in early April 2016 an approach from an eastern direction was chosen, using a multicopter as carrier for gas sampling and sensing instruments. The take-off area was most of time at the northern shelter (see Fig. 1).

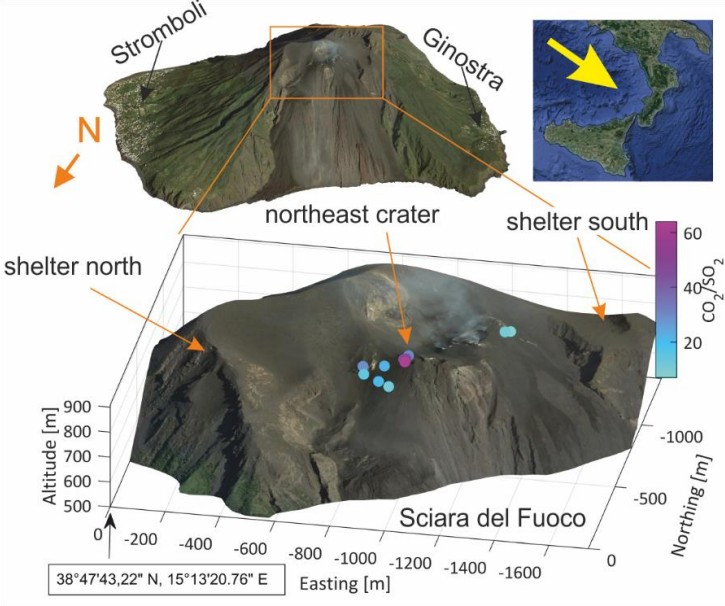

**Figure 1: Overview on the sampling site at Stromboli Volcano with the locations of the retrieved $CO_2/SO_2$ mixing ratios given in Table 2**



## 2. 2 Masaya

Masaya volcano (elevation ~ 600 m a.s.l.), Nicaragua, is a basaltic-andesite shield volcano caldera (6 x 11 km in size) hosting a set of vents. The currently active vent is situated in the Santiago pit crater, formed in 1858-1859 (McBirney, 1956). Masaya persistently emits voluminous quantities of $SO_2$, with fluxes typically ranging from 500 T/d to 2500 T/d (e.g. de Moor et al.,

2013; Carn et al., 2017), making this volcano the single largest contributor to volcanic gas emissions in the Central American Volcanic Arc (Mather et al., 2006). In January 2016 the reoccurrence of a new superficial lava lake (~40 x 40 m) was observed together with an increase in activity. Due to high emission rates and the low-altitude plume, Masaya volcano has a detrimental environmental impact on the downwind areas, diminishing vegetation and potentially affecting human health (Delmelle et al., 2002). Continuous monitoring of the gas emissions is realized by a stationary Multi-GAS (MG) system (through the DCO-

DECADE program) at the crater rim and two scanning DOAS instruments (NOVAC network (Galle et al., 2010)) in the downwind direction. Besides the presence of a strong plume, Masaya volcano provides perfect conditions for field-testing new methods and studying plume chemistry using UAVs: easy accessibility by car, low altitude, and relative stable dominant wind direction (northeast). In July 2016 flights were launched from the caldera bottom marked "flight area" in Fig. 2 (c).

## 2. 3 Turrialba

Turrialba is a stratovolcano with a peak elevation of 3340 m a.s.l. and is located about 35 km east and directly upwind of San José, the capital of Costa Rica. It is the southernmost active volcano of the Central American Volcanic Arc. In the 2000s, an almost 150 years long period of quiescence has ended and since 2010 several vent opening phreatic eruptions henceforth occurred marking an ongoing but erratic phase of unrest (Martini et al., 2010), characterized by variable ash and gas emission intensities (up to 5000 tons/day of $SO_2$ (de Moor et al., 2016a)). The proximity of Turrialba volcano to the densely populated

central valley with Costa Rica's major international airport and the dominant western wind direction is responsible for ash depositions, causing health and air traffic problems. Therefore, the activity of Turrialba is continuously monitored by various systems including permanent Multi-GAS stations to observe short-term precursory changes in the gas composition prior to eruptive events (de Moor et al., 2016a). However, stations located near the active vent suffer from ash deposition during more frequent episodes, making maintenance risky or impossible. Furthermore, the accessibility of the summit and surrounding

areas is degrading due to intense erosion following vegetation destruction by acid rain, heavy rainfall, ash deposition and remobilization, and the lack of infrastructural maintenance following community evacuation. Thus, the use of UAV-based systems might represent the only viable approach for in-situ measurements of the open vent plume during periods of high activity. In 2016 flights were conducted starting at the "La Silva" site (Fig. 2 (d)) to investigate the feasibility of UAV-based gas sensing system at this challenging environment (high altitude and thick ash plumes).



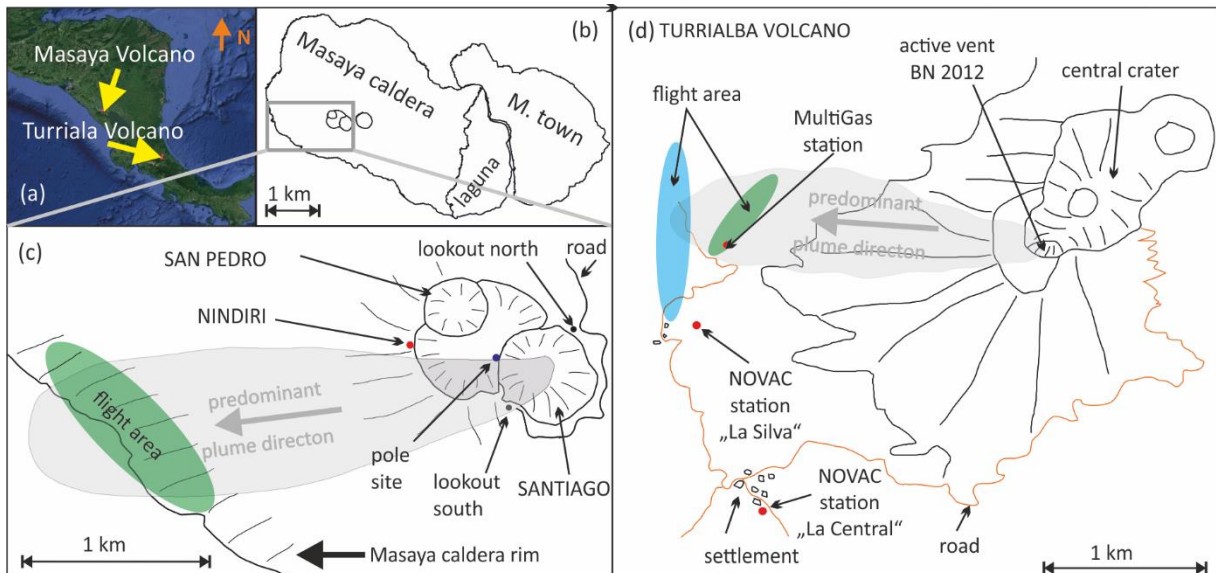

**Figure 2: Overview on the field work sites at (a-c) Masaya volcano (Nicaragua) and (d) Turrialba volcano (Costa Rica). Sampling locations mentioned in Table 3 are marked in (c) as follows: Santiago rim, lookout south (black marker); Santiago rim, pole site (blue marker); Nindiri rim (red marker)**

## 3. Instrumentation

### 3.1 Unmanned aerial vehicle

The UAV (called RAVEN, **r**emote-controlled **a**ircraft for **v**olcanic **e**mission a**n**alysis, see Fig. 3 (a) – (b)) used during the field campaigns was a four-rotor multicopter with foldable arms (Black Snapper, Globe Flight, Germany) with an E800 motor set using propeller with a diameter of 13 inch and a pitch of 4.5 inch (DJI Innovations, Shenzhen, China). It was flown manually in line-of-sight conditions. The multicopter had a weight of 2.3 kg including a 22.2 V (6S) 4.5 Ah battery. A maximum payload of 1.3 kg was achieved with various mounted instruments. The foldable frame of the UAV was beneficial in regards of the usual necessity to personally carry equipment into field, especially on volcanoes like Stromboli. Another advantage of this system was that the battery capacity is within the guidelines of air travel restrictions allowing the system to be transported on commercial airplanes. The main controller (NAZA M-2, DJI Innovations, Shenzhen, China) of the multicopter was connected with a combined denuder sampling and $SO_2$ sensing instrument (hereafter named Black Box (see Sec. 3.3)), to transmit the measured $SO_2$ data as an 0 V - 5 V signal to the remote control, where it is displayed. This allowed the operator to find areas with dense plume to hover the system for stationary sampling and to react to changes of the plume direction, which is challenging if relying only on visual observation of the plume. A data logger (Core 2, Flytrex Aviation, Tel Aviv, Israel) with micro SD card was used to log the flight data from the main controller consisting of GPS coordinates, pressure and temperature data at 2 Hz. The payloads were attached below the main body of the multicopter with an inlet for the in-situ $CO_2$ and $SO_2$ sensing instrument (hereafter called Sunkist (see Sec. 3.2)) close to the center of the copter. The sampled air volume can be



assumed to originate from within a radius of a few meters (see e.g. Roldan et al., 2015; Alvarado et al., 2017; Palomaki et al., 2017), which represents homogeneous conditions for a widely spread out plume. This assumption is confirmed by a self-developed method for the estimation of the origin of sample air, which is described in the supplementary material (Chapter IV).

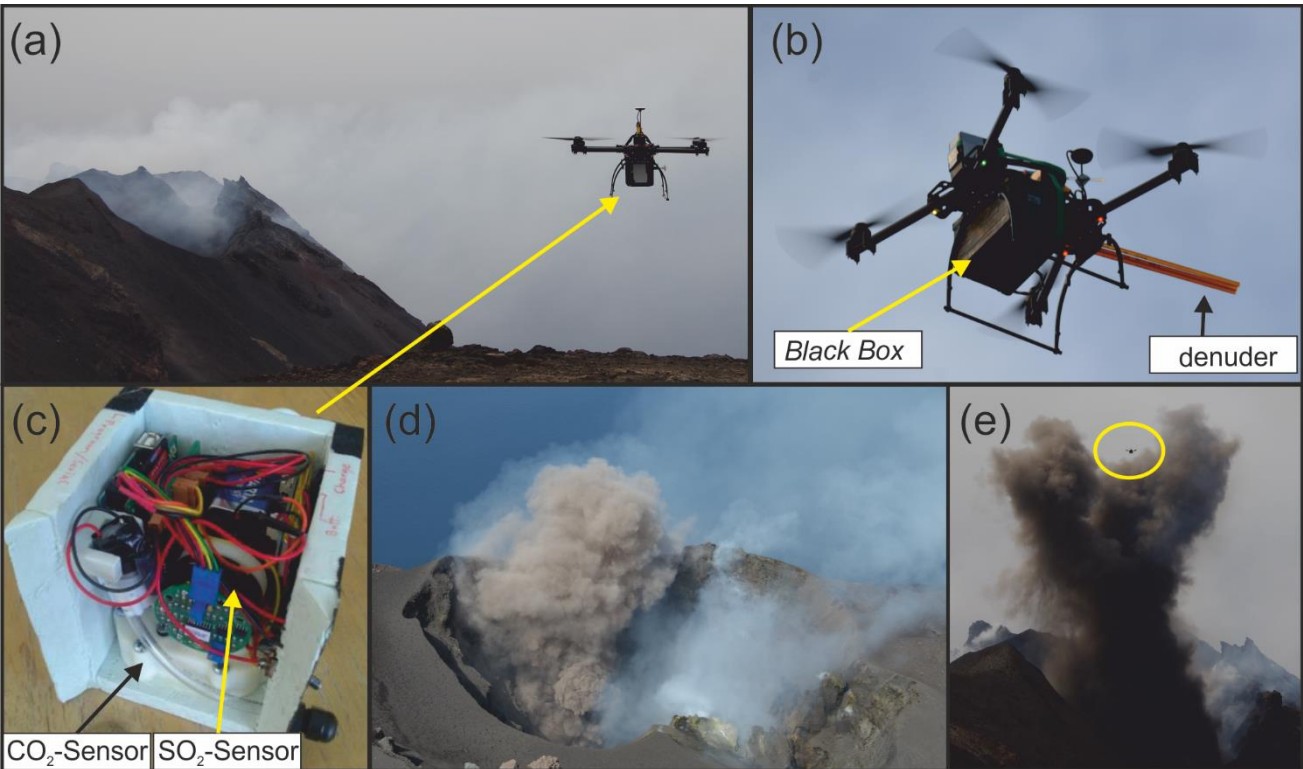

**Figure 3: (a) Sampling site at Stromboli Volcano with Northeast crater in the background and the *RAVEN* UAV carrying the *Sunkist* gas monitor, (b) *RAVEN* UAV with the Black Box sampling unit and three gas diffusion denuders, (c) interior view on the *Sunkist* gas monitor, showing the $CO_2$ and $SO_2$ sensors, (d) passive degassing (white plume) and eruptive ash explosion (brown ash cloud) at the north east crater at Stromboli Volcano, (e) ash eruption at the Northeast crater producing an ascending ash plume with the RAVEN UAV in direct proximity (yellow circle), returning from sampling flight**

### 3.2 $CO_2$/$SO_2$ gas sensors – Multi-GAS (Sunkist)

A gas sensor system was developed for use in volcanic environments with a focus on robustness and compact and lightweight design for application on UAVs. The system, called Sunkist (SK) contains two gas sensors: (1) A $CO_2$ sensor (K30 FR, SenseAir, Delsbo, Sweden), which uses the principle of non-dispersive infra-red absorption spectroscopy: The sample gas gets sucked into a multi reflection cell, where it is exposed to the radiation of a small light bulb. The $CO_2$ concentration is determined by measuring the light attenuation on distinct $CO_2$ absorption bands in the near infra-red. (2) An electrochemical $SO_2$ sensor (CiTiceL 3MST/F, City Technology, Portsmouth, United Kingdom), which basically consists of an electrochemical cell, where oxidation of $SO_2$ on one of the cell's electrodes creates charges and leads to a measureable compensating current between the two cell electrodes.



**Table 1: Components and specifications of the sampling and sensing instruments**

| Component/ Parameter | Specifications | | |
|---|---|---|---|
| | Sunkist | Black Box | |
| CO$_2$ sensor | NDIR, SenseAir CO$_2$ Engine K30 FR | | |
| SO$_2$ sensor | Electrochemical, CiTiceL 3MST/F | Electrochemical, CiTiceL 3MST/F | |
| Operating temperature | 0-50°C (CO$_2$), -20-50°C (SO$_2$) | -20-50°C | |
| Operating humidity | 0-95% (CO$_2$), 15-90% (SO$_2$), non-condensing | 15-90% (SO$_2$), non-condensing | |
| Operating pressure | Atmospheric ± 10% (SO$_2$) | Atmospheric ± 10% | |
| Temperature dependence | 1/T (assumed for CO$_2$), 0.25 %/°C (SO$_2$) | 0.25 %/°C | |
| Pressure dependence | 1.6 %/kPa (CO$_2$), 0.0015 %/kPa (SO$_2$) | 0.0015 %/kPa | |
| Response time | 2 s (CO$_2$), <20 s (SO$_2$) | <20 s | |
| Accuracy | + 5 % signal (CO$_2$), ± 1 % signal (SO$_2$) | ± 1 % signal | |
| Resolution | 0.5 ppm (SO$_2$), 5 ppm (CO$_2$) | 0.5 ppm | |
| Range | 0-5000 ppm (CO$_2$), 0-200 ppm (SO$_2$) | 0-100 ppm | |
| Sampling rate | 2 Hz | 2 Hz | |
| Computer | Arduino, with microSD card logger | Arduino, with SD card data logger and motor-shield to power the pump | |
| Voltage | 9 V for SO$_2$ sensor (alkaline battery), 3.7 V LiPo battery for Arduino | 9 V for SO$_2$ sensor (alkaline battery), 11.1 V LiPo battery for Arduino and pump | |
| Inlet | particle filter | 3 denuders (50 cm) or 3 x 2 denuders (15 cm) | |
| Others | Warm up time | 1 min (CO$_2$) | Micro solenoid valves | First Sensor, Germany TN2P006LM05LB |
| | Additional Sensors | Temperature, pressure, relative humidity | Mass flow meter | First Sensor, Germany, WBAL001DUH0 |
| | | | Micro pump | TCS micropumps, UK, DS250BL |
| Weight | 500 g | 500 g | |
| Dimensions | 14x13x14 cm (LxWxH) | 20x13x14 cm (LxWxH) | |




The $CO_2$ sensor is placed inside a hermetic box, which is part of the air path (Fig. 4 (a)). It is equipped with further on-chip sensors for humidity (SHT21 from Sensirion, Staefa, Switzerland), temperature and pressure (BMP180 by Bosch Sensortec, Reutlingen, Germany), which allow to correct the gas data for dependencies on named environmental parameters.

The sensors are read out by a custom-built Arduino Uno Rev 3 computer with a micro SD card logger at a sampling rate of

2 Hz and powered by a rechargeable 3.7 V lithium polymer (LiPo) battery (Fig. 4 (b)). The $SO_2$ sensor was powered by a separate 9 V alkaline battery. The whole system is sheltered in a polystyrene foam case with a total weight of 500 g (Fig. 3 (c)). A 45-µm pore size PTFE filter was attached to the inlet to prevent particles from entering the sensors. Gas was pumped through to the in-series connected sensors (1. $SO_2$, 2. $CO_2$) by a small pump using a flow rate of 500 ml/min. The system was calibrated before and after field deployments with $CO_2$ (0-1500 ppm) and $SO_2$ (0-30 ppm) test gases. Detailed information on the

specifications of the sensors is shown in Table 1 and in the supplementary material (Fig. S1).

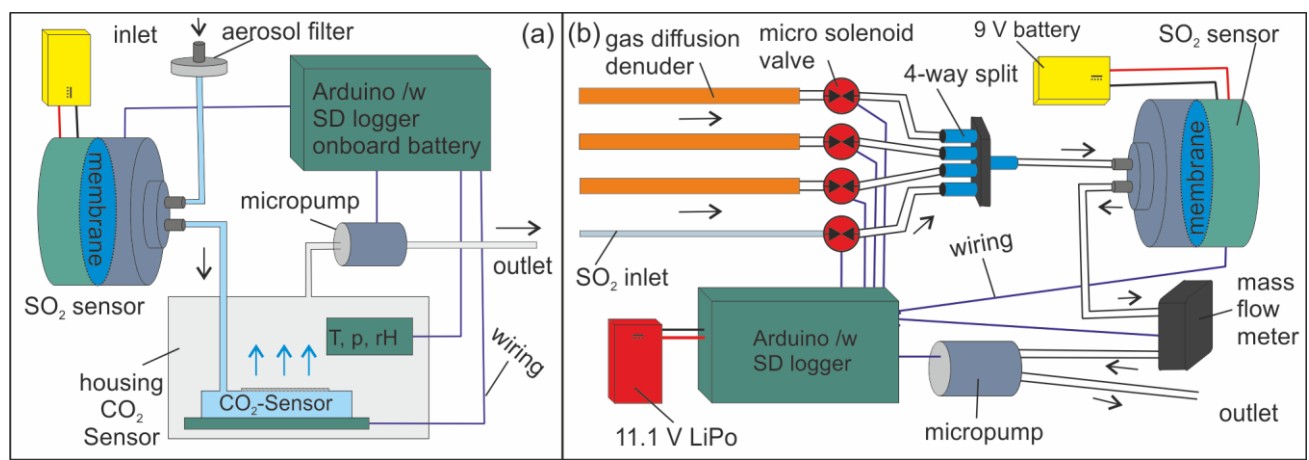

**Figure 4: Schematics of the (a) *Sunkist* monitoring unit and the (b) *Black Box* gas sampling system**

### 3.3 Gas diffusion denuder sampler (Black Box)

An in-situ gas sampling system was constructed to enable gas diffusion denuder sampling in the plume at various distances from the vent using the UAV. To compensate for dilution, a CiTiceL 3MST/F $SO_2$ sensor (City Technology, Portsmouth, United Kingdom) was implemented to obtain halogen/sulfur ratios combining denuder samples and sensor data. The sampler (called Black Box (BB)) consisted of the following components, which are introduced in the order the gas passes through: I) inlet system for three denuders; II) electrochemical $SO_2$ sensor; III) mass flow sensor; IV) micro gas pump (see Figure 4 (b)

and Table 1). The housing was made of polystyrene foam to ensure weight requirements. An Arduino microcontroller (Uno Rev 3) for signal processing and data logging on a SD card was built in. Various 11.1 V LiPo batteries (500 and 1000 mAh) supplied the power, with different capacities depending on the desired payload and operation time. The Arduino computer transmitted a pulse width modulated signal between 0 V and 5 V, proportional to the detected $SO_2$ mixing ration, to the main controller of the multicopter via cable connection, which then was sent by telemetry to the remote control to allow the operator



to assess plume strength in real time to optimize denuder exposure time. The $SO_2$ gas sensor was calibrated in the laboratory and close to field conditions with $SO_2$ gas standards in $N_2$ (0 – 54.1 ppm).

Two types of coating materials were used as derivatization agent for the gas diffusion sampling. Total reactive bromine species (BrX) were determined by denuders coated with 15 µmol of 1,3,5-trimethoxybenzene (TMB) - which reacts to 1-bromo-2,4,6-

trimethoxybenzene - and subsequent gas chromatography-mass spectrometry (GC-MS) analysis (Rüdiger et al., 2017). Hydrogen bromide (HBr) was sampled with denuders coated with 7.2 µmol of 5,6-epoxy-5,6-dihydro-1,10-phenanthroline (EP) - which is selectively reactive towards halogen acids through its epoxy function forming 5-halogeno-6-hyrdroxy-5,6-dihydro-1,10-phenanthroline - and analyzed by liquid chromatography-mass spectrometry (LC-MS). For the UAV-based application 15 cm long denuders were used at a sampling flow rate of 208 ml/min to ensure quantitative sampling. Denuders

with both coating types were sampled at the same flow rate simultaneously to the recording of Multi-GAS (Sunkist) data.

### 3.4 Drone-operated miniature differential optical absorption spectroscopy (DROAS)

A miniature UV spectrometer system (Galle et al., 2003) was employed for flying mobile DOAS traverse measurements to conduct estimations of the $SO_2$ flux at Turrialba volcano. This system consisted (Fig. 8 (a)) of an UV spectrometer (USB2000+, Ocean Optics, USA), a miniature telescope (Ocean Optics 74-DA collimating lens, diameter: 5 mm, focal length 10 mm), a

GPS Antenna (BU-353-S4, GlobalSat, Taipei, Taiwan) and a miniature on-board computer (VivoStick TS10, ASUS, Taipei, Taiwan). The system was powered by a 11.1 V LiPo battery (1000 mAh), which was connected via a switching regulator (CC BEC 10 A, Castle Creations, USA) to give 9 V at 2 A. The spectrometer and the GPS antenna were connected (and powered) at the computer via USB ports. The NOVAC software "mobile DOAS" developed at the Chalmers University (Sweden) was run on the computer for data acquisition and later evaluation. The miniature PC in the spectrometer system was accessed via

a remote desktop connection by a different computer to initialize the data acquisition by the "mobile DOAS" software. Evaluation of the $SO_2$ fluxes obtained by DROAS traverses was achieved by a comparison with the $SO_2$ fluxes derived by two NOVAC stationary DOAS instruments (*La Silva* and *La Central*, see Fig. 2 (d)) located in proximity to the flight area.

### 3.5. Data Processing

### 3.5.1 Sensor calibration

All three in-situ gas sensors (two identical $SO_2$, one $CO_2$) were calibrated using test gas standards mixed with nitrogen, using either tedlar bags, dynamic dilution or readily mixed test gases. The sensors were exposed to different mixing ratios by pumping the gas mixes through the system. Calibration functions were fitted, including errors in mixing ratio and signal (York et al., 2004) to give sensitivities (slope) and offset levels (intercept) (Supplementary Material Fig. S2 – S4). As the $CO_2$ sensor responds to the gas concentration (molecules per volume), $CO_2$ mixing ratios (molecules per molecules of air) were obtained

by compensating the concentration signals with pressure and temperature data recorded by the built in sensors, assuming ideal gas behavior. The $SO_2$ sensor output however, relates directly to the mixing ratio, due to the diffusivity of the transport





membrane being inversely proportional to the pressure and therefore cancelling out the pressure dependency of the concentration. The two gas sensors operate with significantly different response times ($T_{90}$ for $CO_2$ ~ 2 s, for $SO_2$ ~ <20 s), since sample gas enters the $SO_2$ sensor only by molecular diffusion, whereas in the $CO_2$ sensor it is directly driven through the optical cell. To adapt the response times of the two sensors, the $CO_2$ signal was smoothed through convolution with a first

order transfer function. The transfer function's response time factor was chosen, such that the correlation of $CO_2$ and $SO_2$ signal got maximized for discrete peaks (see Fig. 5 (a)).

### 3.5.2 Gas diffusion denuder analysis

The sampled gas diffusion denuders were sealed air tight and stored in darkness for subsequent analysis. The coating, which contained the derivate and derivatization agent in excess (TMB or EP), was eluted off the denuders using 5 times 2 mL of 1:1

of ethyl acetate and ethanol (in case of TMB) or 5 times 2 mL methanol (EP). After the elution, the solvent was evaporated (at 35 °C under gentle $N_2$ gas stream) to a volume of approximately 100 μL. Internal standards (with TMB: 2,4,6-tribromoaniline; with EP: neocuproine) were added to each sample solution to account for evaporation losses. The condensed samples were analyzed by GC-MS (TMB) and LC-MS (EP) and quantified using external calibration. Due to the lack of a pure calibration standard hydrogen bromide was only measured qualitatively. Halogen mixing ratios in air were derived from the measured

halogen amounts on the denuders and the respective sampling volume obtained by the sampling pump data. In addition to the actual sample denuders open field blank denuders were prepared to account for potential diffusive gas precipitation during the flights.

### 3.5.3 Gas ratios

For a feasible data interpretation, gas ratios were calculated from the sensor data and the denuder analysis results. Halogen

mixing ratios were interpreted concerning dilution with ambient air by relating to $SO_2$, which is rather slow in its oxidation, and therefore can be treated as a stable dilution proxy for a short term plume observation (Porter et al., 2002). Thus, the derived halogen mixing ratio was divided by the time-integrated $SO_2$ mixing ratios obtained during the denuder sampling period to obtain $BrX/SO_2$ ratios. Due to a baseline drift in the $CO_2$ sensor data, which was only observable during long-term measurements (> 45 minutes), the pressure and time-response corrected $CO_2$ data was additionally drift corrected for long-

term measurements. To do so, a linear fit was made to the sloped background signal and subtracted from the $CO_2$ signal (see supplementary material Fig. S5). $CO_2/SO_2$ ratios were calculated with a linear fit to $CO_2$ vs. $SO_2$ scatter plots, considering the deviations (York et al., 2004) of the two sensor signals.

### 3.5.4 DROAS evaluation and gas fluxes

$SO_2$ emission rates during the flights at Turrialba were derived by traversing the plume with the DROAS instrument pointing

vertically upwards in the direction of the plume. The "mobile DOAS" software developed at Chalmers University was used to control the spectrometer and to retrieve $SO_2$ column amounts from the spectra. The $SO_2$ columns were achieved using a



wavelength evaluation window of 310-330 nm and including $O_3$ and $SO_2$ absorption cross sections convoluted with a slit function as well as a Ring Spectrum and a 3rd order polynomial in the DOAS fitting routine. The calculation of the $SO_2$ emission rate involves the integration of the $SO_2$ column amounts measured along the flight path resulting in a cross-sectional $SO_2$ area, which was geometrically corrected to obtain a surface that is orthogonal to the plume direction, and then multiplied by the wind speed (obtained from the NOAA National Center for Environmental Predictions (NCEP) Global Forecast System) to calculate $SO_2$ fluxes. Further details on the spectral evaluation routines and flux calculations can be found elsewhere (de Moor et al., 2017).

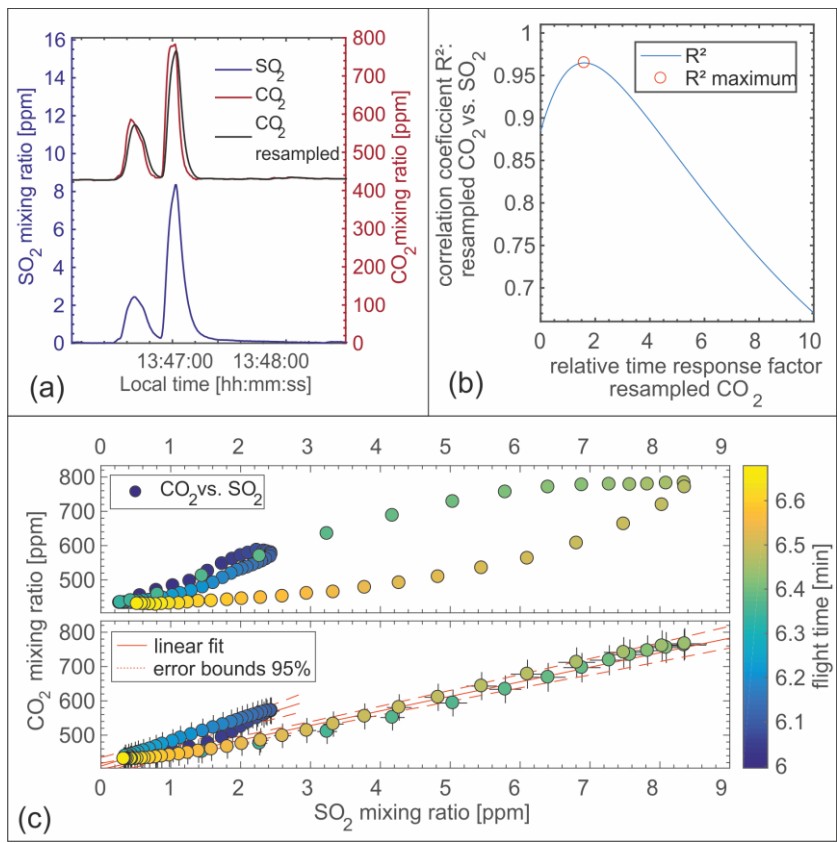

**Figure 5: (a) Example of time series for mixing ratios of $SO_2$ and $CO_2$ (original data in red, resampled data in black), showing discrete gas masses at Stromboli volcano (1st flight on 05th April 2016), (b) Correlation plot for the determination of the relative time response factor for the $CO_2$ gas sensor with a maximum at a relative time response factor of 1.7, (c) $CO_2$ over $SO_2$ mixing ratios, showing the outcome of the resampling of the fast $CO_2$ with a relative time response factor of 1.7 (lower plot), linear regression results $CO_2/SO_2$ ratios of 64 ± 16 the first peak and 42 ± 4 for the second.**



# 4. Results and discussion

## 4.1. Multicopter performance assessment

During the deployment at three volcanoes, the RAVEN multicopter conducted more than 50 flights under moderate wind conditions, in most cases below 10 m/s. The multicopter achieved a maximum operation altitude of 3320 m at Turrialba volcano

and records showed a maximum speed of 85 km/h. With a takeoff weight of 2.45 kg and a payload of maximum 1.3 kg flights at Turrialba volcano were still possible, although the flight time was reduced to about 5 to 8 minutes. At the Masaya volcano sites (takeoff altitude ~ 500 m), a maximum ascent above ground level 1080 m was recorded. A typical flight time at this site was between 10 and 15 minutes. The telemetrically transmitted $SO_2$ mixing ratios from the Black Box apparatus allowed the localization of high plume densities and therefore adjustments on the optimal hover location. While flying in the line of sight

the system could be reliably controlled within a distance of 1-2 km to the operator. However, entering the dense plume proved to challenge the connection between remote control and receiver, resulting in multiple connection losses during flights, in which the UAV also left the line-of-sight of the operator. Nevertheless, GPS connection was still present and an automated return mechanism allowed regaining control after the UAV left areas of high plume densities. At Masaya this phenomenon was observed mostly in a plume with condensed water and $SO_2$ mixing ratios in low one-digit ppmv numbers, while in a plume

without condensation close to the crater control was maintained even with $SO_2$ levels up to 40 ppmv. This is probably due to the attenuation effect of fog and cloud droplets on millimeter-waves, as they are used with the 2.4 Ghz transmitter of the remote control (Zhao and Wu, 2000).

## 4.2 $CO_2$/$SO_2$ gas sensors

During two field deployments at Stromboli (Table 2) and Masaya (Table 3) volcanoes, the lightweight gas monitoring system

Sunkist (SK) determined $CO_2$/$SO_2$ gas ratios at various airborne and ground-based locations. A comparison with a stationary Multi-GAS (MG) instrument at Masaya volcano for the same site and time (lookout point south, 14th July2016, 11:19) and with inlets of both systems in proximity to each other gave results on $CO_2$/$SO_2$ ratios that were within each other's 2-sigma confidence intervals ($CO_2$/$SO_2$ of 2.9 ± 0.3 for MG and 3.6 ± 0.4 for SK). The time series of the $SO_2$ mixing ratios of SK and MG also showed a good agreement ($R^2$ = 0.89) (Fig. 6). An application of SK further downwind (0.5 km from the rim) gave a

$CO_2$/$SO_2$ ratio of 3.3 ± 1.2, while the MG measured a $CO_2$/$SO_2$ ratio of 3.1 ± 0.1 during the same time at the rim. This is showing SK's ability for deployment in a more diluted plume, although with the disadvantage of higher errors, due to the higher relative background in $CO_2$ at more distant locations. Furthermore, UAV-based application (9 flights, 17th – 20th July 2017) between 1.5 km and 2 km downwind of the Masaya plume resulted in average 42 % higher $CO_2$/$SO_2$ ratios with a larger standard deviation compared to the crater rim (14th – 16th July 2017), but still within each other's errors ($CO_2$/$SO_2$: 5.4 ± 2.3

at 1.5 - 2 km; 3.8 ± 0.3 at the rim). Due to the limitations of the $CO_2$ sensing, acquirement of useful $CO_2$/$SO_2$ data with SK is more feasible in dense plumes close to, but not limited to the crater rim, as the airborne application has shown. Additionally, $SO_2$ mixing ratios were measured as a plume dilution proxy by the Black Box (BB) system. Both the BB and SK systems use



an identical $SO_2$ sensor and showed a good agreement of their $SO_2$ time series and time integrated $SO_2$ mixing ratio (SK: 1.75 ± 0.08 ppmv, BB: 1,84 ± 0.08 ppmv) (supplement material S6).

At Stromboli volcano, the Sunkist and Black Box systems were deployed on two days (05[th] & 06[th] April 2016), resulting in seven flights into the plume. These flights covered distances, between 11 m and 419 m from the vent in a downwind direction (northeast), above the Sciara del Fuoco. As shown in Figure 1, discrete gas masses with different $CO_2/SO_2$ compositions were measured. The retrieved $CO_2/SO_2$ ratios ranged between 7 and 64, with the higher values typically detected directly above the vent (11 m to 26 m) (see Tab. 3 and Fig. 1). During the multicopter operations close to the vent regular strombolian ash explosion occurred (see Fig. 3 (d) and (e)), which are likely accompanied by $CO_2$-rich gas masses (La Spina et al., 2013) and therefore a possible explanation for the detected high $CO_2/SO_2$ in a rather undiluted plume region.

On both flight days the predominant wind direction was southwest (207° ± 15° and 210° ± 18°, data from weather station at 77 m a.s.l., commercially available from windfinder.com, Kiel, Germany), which resulted in the plume mostly being present across the Sciara del Fuoco and therefore only accessible by UAV. Nevertheless, a local Multi-GAS station (placed on the SE rim of the crater terrace) has discontinuously measured the plume during the period of the UAV survey, showing $CO_2/SO_2$ ratios between 2.2 and 13.6, which is in agreement with some of the multicopter-based measurements. Similar $CO_2/SO_2$ ratios have been observed in the past and are exemplary for an ordinary Strombolian activity (Aiuppa et al., 2009). It has to be taken into account that both instruments have not measured simultaneous or in proximity to each other. Furthermore, a comparison of SK and MG $CO_2/SO_2$ data might be more accurate with the high SK $CO_2/SO_2$ values (associated with eruptive degassing) left aside, as we lack the observations on passive, respectively eruptive degassing behavior for the MG data records.

**Table 2: Overview on the retrieved CO2/SO2 ratios with parameters for the linear fit at Stromboli volcano**

| Date | Time | Flight / Peak | | $CO_2/SO_2$ | lower $SO_2$ limit /ppmv | Data Points | max. $SO_2$ /ppmv | estimated distance /m |
|------|------|------|------|-------------|---------------------------|-------------|-------------------|------------------------|
| 05.04 | 13:46 | 1 / | 1 | 64 ± 16 | 1 | 39 | 2.6 | 26 |
| 05.04 | 13:46 | 1 / | 2 | 42 ± 4 | 1 | 48 | 8.5 | 11 |
| 05.04 | 14:31 | 2 / | 1 | 43 ± 8 | 1 | 31 | 4.5 | 18 |
| 05.04 | 14:32 | 2 / | 2 | 31 ± 12 | 1 | 29 | 4.7 | 25 |
| 05.04 | 16:18 | 3 / | 1 | 7 ± 5 | 0.1 | 101 | 5.8 | 419 |
| 05.04 | 16:23 | 3 / | 2 | 11 ± 11 | 0.1 | 133 | 1.9 | 399 |
| 06.04 | 13:03 | 4 / | 1 | 27 ± 25 | 0.2 | 326 | 0.9 | 155 |
| 06.04 | 13:44 | 5 / | 1 | 22 ± 5 | 0.5 | 324 | 5.2 | 80 |
| 06.04 | 14:44 | 6 / | 1 | 10 ± 13 | 0.2 | 111 | 1.6 | 170 |
| 06.04 | 14:45 | 6 / | 2 | 21 ± 7 | 0.5 | 415 | 2.3 | 177 |
| 06.04 | 15:10 | 7 / | 1 | 9 ± 5 | 0.2 | 516 | 1.6 | 167 |
| 06.04 | 15:16 | 7 / | 2 | 21 ± 5 | 0.5 | 35 | 2.3 | 107 |





**Table 3: $CO_2/SO_2$ mixing ratios and calculation parameters obtained from the Sunkist (SK) and Multi-GAS (MG) instrument at Masaya Volcano. The exact locations are given in Fig 2, errors indicate 2 sigma interval (95.5 %) retrieved by a linear regression (York et al., 2004) including the measurement errors of both the $SO_2$ (error: 5-10 %) and $CO_2$ (error: 5.5 %) sensor**

| Date | Time | Instru-ment | Location / Flight | $CO_2/SO_2$ | lower $SO_2$ limit /ppmv | Data Points | max. $SO_2$ | distance to rim /km |
|---|---|---|---|---|---|---|---|---|
| 14.07.2016 | 09:43 - 15:10 | MG | Santiago rim; lookout south | 2.9 ± 0.1 | 4 | 3200 | 31.1 | 0 |
| 14.07.2016 | 11:18 - 11:44 | MG | Santiago rim; lookout south | 2.9 ± 0.3 | 4 | 440 | 17.5 | 0 |
| 14.07.2016 | 11:19 - 11:45 | SK | Santiago rim; lookout south | 3.8 ± 0.5 | 4 | 1460 | 17.9 | 0 |
| 14.07.2016 | 10:42 - 12:09 | SK | Santiago rim; lookout south | 3.6 ± 0.4 | 4 | 2600 | 18.5 | 0 |
| 15.07.2016 | 09:43 - 16:23 | MG | Santiago rim; pole site | 3.1 ± 0.1 | 4 | 2950 | 29.6 | 0 |
| 15.07.2016 | 15:28 - 16:19 | SK | Nindiri rim | 3.3 ± 1.2 | 1 | 3100 | 5.8 | 0.5 |
| 16.07.2016 | 11:18 - 12:19 | SK | Santiago rim; pole site | 4.7 ± 0.3 | 4 | 3000 | 28 | 0 |
| 17.07.2016 | 08:56 - 08:59 | SK | caldera valley; flight #A2 | 2.9 ± 13 | 1 | 150 | 2.8 | 1.8 |
| 17.07.2016 | 11:21 - 11:28 | SK | caldera valley; flight #A5 | 4.8 ± 3.2 | 1 | 630 | 4.8 | 1.8 |
| 17.07.2016 | 11:53 - 12:01 | SK | caldera valley; flight #A6 | 6.5 ± 3.4 | 0.2 | 620 | 3.4 | 1.9 |
| 17.07.2016 | 12:10 - 12:20 | SK | caldera valley; flight #A7 | 6.2 ± 5.6 | 0.4 | 1150 | 2.2 | 1.5 |
| 18.07.2016 | 07:55 - 09:36 | SK | Santiago rim; pole site | 4.1 ± 0.3 | 4 | 4350 | 36.7 | 0 |
| 18.07.2016 | 13:19 - 13:27 | SK | caldera valley; flight #B4 | 5.1 ± 4 | 1 | 560 | 4.4 | 1.7 |
| 20.07.2016 | 12:55 - 13:04 | SK | caldera valley; flight #C1 | 7.5 ± 5.8 | 0.5 | 310 | 2.8 | 1.9 |
| 20.07.2016 | 13:29 - 13:35 | SK | caldera valley; flight #C2 | 7.7 ± 5.1 | 0.75 | 470 | 2.9 | 1.9 |
| 20.07.2016 | 13:58 - 14:07 | SK | caldera valley; flight #C3 | 2.2 ± 11 | 0.5 | 380 | 1.9 | 1.9 |
| 20.07.2016 | 16:22 - 16:29 | SK | caldera valley; flight #C7 | 5.3 ± 4.4 | 0.5 | 550 | 3.4 | 2 |





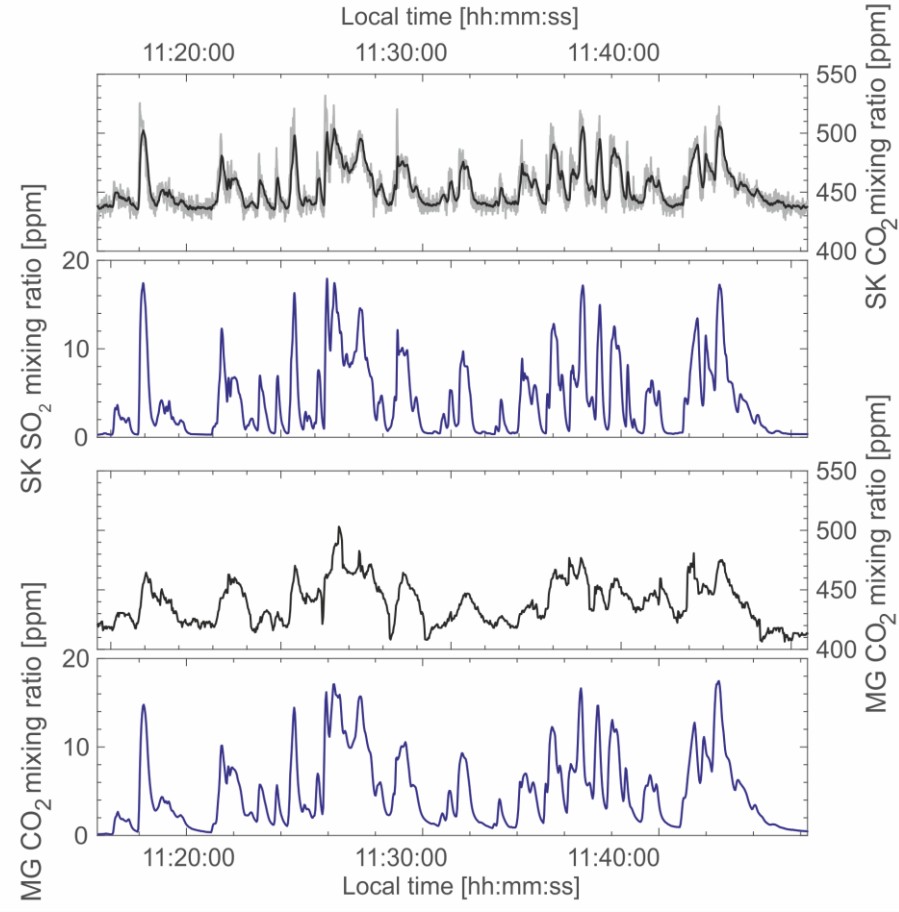

**Figure 6: Comparison of SO₂ and CO₂ time series of a Multi-GAS (MG) instrument and the *Sunkist* (SK) unit at the Masaya volcano crater rim (for SK SO₂ raw data in grey, resampled data in black), both instruments inlets were place in proximity to each other (14th July 2016); SK CO₂/SO₂ = 3.63+/- 0.43 background CO₂ = 439 ppm); MG CO₂/SO₂ = 2.94 +/- 0.30 (background CO₂ = 413 ppm); (additional scatter plots in the supplementary material)**

### 4.3 Halogen measurements

Bromine species were detected by gas diffusion denuder sampling on three of the seven flights at Stromboli volcano. Reactive bromine species were measured between 0.14 and 0.65 ppb (see Table 4), while HBr was determined qualitatively in all three samples.

10    The obtained ratios of reactive bromine (BrX) to SO₂ ($1.9*10^{-4}$ – $9.8*10^{-4}$) are within the range of bromine to sulfur ratios produced by other methods at Stromboli volcano, e.g. alkaline trap sampling by Wittmer et al. (Wittmer et al., 2014) ($4.3*10^{-4}$ – $2.36*10^{-2}$). Figure 7 (b) shows the BrX/SO₂ ratios for different plume ages, which was calculated by taking wind speeds into account. Although the general feasibility of the used methods for the investigation of reactive halogen species in an aging plume is demonstrated in this proof-of-principle approach, a trend in the BrX/SO₂ ratio over age is not recognizable

15    for this limited data set. Without information on abundances of other halogen species such as BrO(g), HBr(g) and aqueous





particulate bromine ($HBr_{(aq)}$), interpretation of the $BrX/SO_2$ ratio is difficult as the emitted gas composition may also change on shorter time scales (La Spina et al., 2013) compared to the campaign duration. However, an increase of the reactive bromine species BrO with distance from the crater rim has been observed previously by DOAS measurements at various volcanoes and is well described in the literature (Oppenheimer et al., 2006; Bobrowski et al., 2007). Although the bromine

species in volcanic plumes has been subject of several ground-based (e.g. Gliß et al., 2015) airplane (e.g. General et al., 2015), satellite (e.g. Theys et al., 2009; Hörmann et al., 2013) and model studies (e.g. Bobrowski et al., 2007; Roberts et al., 2009; von Glasow, 2010; Roberts et al., 2014; Jourdain et al., 2016) in recent years, in-situ measurements are scarce. The here presented data for the first minute after emission highlights the potential of UAV-based measurements to improve sample acquisition and thus a better understanding of plume aging.

As shown in Figure 7 (c), the BrX to $SO_2$ ratio seems to change not only with the plume age but also with $CO_2/SO_2$ mixing ratios, which were simultaneously measured. However, with only three data points a further interpretation is inadequate due to lack of a statistical basis. Nevertheless, these first results show the principal practicality of the used denuder sampling and gas sensing methods for simultaneous investigation of halogen, carbon and sulfur emissions.

**Table 4: Sample parameter and bromine measurement results for three denuder samples at Stromboli volcano.**

| Sample number | 1 | 2 | 3 |
|---|---|---|---|
| Date | 05.04.2016 | 06.04.2016 | 06.04.2016 |
| Time | 14:31 | 13:46 | 14:45 |
| Duration /s | 53 | 364 | 320 |
| Sample volume /L | 0.18 | 1.26 | 1.11 |
| pressure /mbar | 906 | 914 | 914 |
| integrated $SO_2$ /ppmv | 1.85 ± 0.09 | 0.34 ± 0.04 | 0.74 ± 0.05 |
| BrX /ppb | 0.65 ± 0.06 | 0.34 ± 0.01 | 0.14 ± 0.1 |
| $BrX/SO_2$ *$10^{-4}$ | 3.5 ± 0.4 | 9.8 ± 1.3 | 1.9 ± 0.2 |
| $CO_2/SO_2$ | 43 ± 9 | 27 ± 4 | 17 ± 8 |
| distance /m | 21 ± 2 | 80 ± 2 | 177 ± 2 |
| wind speed /m s$^{-1}$ | 3.4 ± 0.5 | 4.8 ± 0.5 | 4.8 ± 0.5 |
| plume age /s | 6 ± 1 | 17 ± 2 | 37 ± 4 |



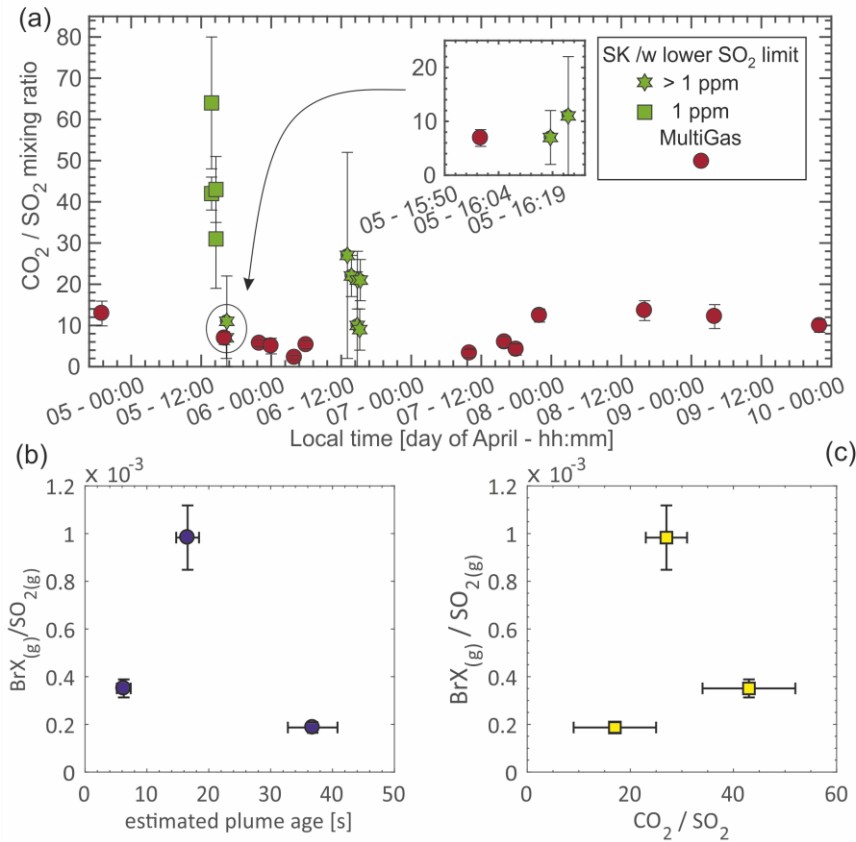

**Figure 7: (a) Measured CO₂/SO₂ gas ratios over a 5-day period by a Multi-GAS station (located east of the crater terrace) and airborne measurements with the *Sunkist* system, data point shapes indicate the lower SO₂ mixing ratio limits for the linear fit over the CO₂ vs SO₂ data, (b) development of the gaseous reactive bromine species over SO₂ over the estimated plume age (derived from distance to crater and estimated wind speed), (c) gaseous reactive bromine/SO₂ ratios vs. CO₂/SO₂ ratio**

### 4.4. DROAS measurements

On September 27th, two DROAS plume transects were performed at Turrialba volcano during mild ash emission at around 2800 m a.s.l. with a total flight time of approximately 10 minutes. The flight was conducted in the downwind direction west of the active vent and crossed underneath the plume on a south-north axis (Fig. 8 (b)), exiting the plume on both ends of the flight path, which is indicated by the low $SO_2$ column amounts close to the baseline in the northern- and southern-most section of the flightpath (Fig. 8 (c)). $SO_2$ fluxes were calculated for that flight and resulted in $1776 \pm 1108$ T/d $SO_2$ for traverse A and $1616 \pm 1007$ T/d $SO_2$ for traverse B (see Table 5). Calculation of the $SO_2$ fluxes obtained by the two scanning DOAS instrument from the NOVAC network, located on the southern edge of the plume, gave average results of $1533 \pm 986$ T/d $SO_2$ for *La Silva* and $1094 \pm 704$ T/d $SO_2$ from *La Cetral* for a 2-hour period, in which the flight took place. Therefore, the results of the DROAS traverses are in a good agreement with the scanning DOAS stations. De Moor et al. (de Moor et al., 2016a) conducted car based mobile DOAS transects, which typically take ~45 minutes for a round trip due to rough terrain. Dynamic





gas plumes can significantly change travel direction on scales of minutes, introducing a significant source of error to car traverses. A major advantage of the UAV method is that it is much quicker, about 10 minutes for a round trip, thus providing a more accurate snapshot of the plume.

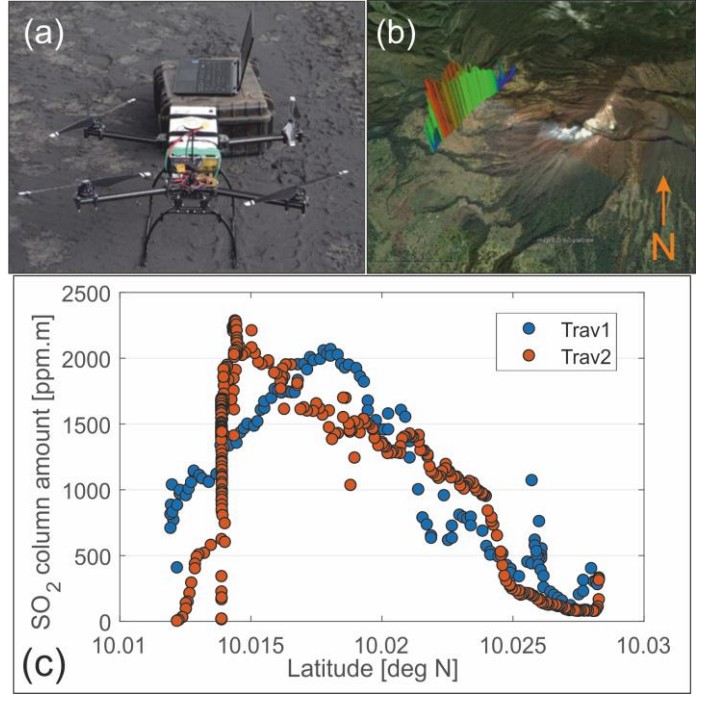

**Figure 8: (a) Drone-operated miniature DOAS setup, (b) Illustration of SO₂ column amounts measured by DROAS at Turrialba volcano, (c) SO₂ column amounts during the transversal flight underneath the plume**

**Table 5: SO₂ fluxes obtained by DROAS traverses and two stationary scanning DOAS instruments at Turrilaba volcano. The average fluxes from La Silva and La Central were derived for a 2-hour period in which the DROAS flights were conducted, maximum fluxes for this period as presented as well**

| Instrument/Traverse | SO₂ flux at 1 m/s | | | error | windspeed | | | SO₂ flux | | |
|---|---|---|---|---|---|---|---|---|---|---|
| [unit] | [T/d] | | | [%] | [m/s] | | | [T/d] | | |
| **DROAS Traverse A** | 419.9 | ± | 38.6 | 9.2 | 3.78 | ± | 2.62 | **1776** | **±** | **1108** |
| **DROAS Traverse B** | 381.6 | ± | 35.5 | 9.3 | 3.78 | ± | 2.62 | **1614** | **±** | **1007** |
| **La Silvia NOVAC station AVERAGE** | 362.3 | ± | 72.5 | 20 | 3.78 | ± | 2.62 | **1533** | **±** | **986** |
| **La Silvia NOVAC station MAX** | 481.72 | ± | 96.3 | 20 | 3.78 | ± | 2.62 | **2038** | **±** | **1312** |
| **La Central NOVAC station AVERAGE** | 258.6 | ± | 51.7 | 20 | 3.78 | ± | 2.62 | **1094** | **±** | **704** |
| **La Central NOVAC station MAX** | 448.1 | ± | 89.6 | 20 | 3.78 | ± | 2.62 | **1896** | **±** | **1220** |



## 5. Conclusion and outlook

In this study we have demonstrated the feasibility of a multicopter-based approach for volcanic in-situ plume measurements to achieve investigations on the compositional variability of an aging plume. The use of a multicopter UAV proved to be a suitable alternative to ground-based operations, especially in hard to access or not at all accessible areas, like active volcanic vents or elevated downwind plumes. The aerial sampling systems demonstrated robustness and effectiveness during the field missions with harsh environmental and meteorological conditions, including ash-laden plumes. With the newly developed sampling system, reactive halogen species were observable in previously inaccessible downwind plume areas. Halogen speciation information enhances our understanding of plume chemistry in the aging plume, which represents important knowledge for new volcano monitoring approaches. Therefore, in-situ speciation methods for halogens should be extended and optimized for other gaseous and aqueous compounds including chlorine and iodine species. Additionally, at Stromboli volcano changes in the $BrX/SO_2$ ratios were observed with different $CO_2/SO_2$ ratios, which represents an interesting matter and should be further explored in future studies.

Furthermore, multicopter-based $CO_2/SO_2$ ratio measurements showed its reliability, opening a promising approach for monitoring inaccessible volcanoes or during dangerous eruptions minimizing personnel risk at highly active volcanoes. Although we demonstrated the applicability of the Sunkist lightweight gas sensing system for potential UAV based monitoring, challenges in the acquisition of high-quality data for diluted plumes still exist. Thus, high-class components (e.g. sensors, microcontroller or data logger) promise to achieve better sensitivities, but with the disadvantage of higher losses in the case of a crash.

Moreover, the herein presented UAV application of a miniature DOAS instrument (DROAS) for $SO_2$ gas flux measurements has shown its quality in the harsh environment of Turrialba volcano. The potential to be an excellent alternative to walking or car-based traverses for $SO_2$ flux acquisition has been proven with a significant reduction of risks and operation time.

The case studies presented here explored the general feasibility of multirotor UAVs in volcanic plume studies and are an initial step in the direction of remotely operated gas measurements at active volcanoes. We have shown the potential of UAV-based sampling to gain insight into reactive halogen chemistry and processes of plume aging, and further application of this method will yield data from previously unstudied plume regions. Technological advances promise to enable autonomous UAV operations, with extended flight times, such as scheduled pre-programmed flight paths through volcanic gas plumes for hazard assessment.

## Competing interests

The authors declare that they have no conflict of interest.





**Acknowledgements**

Julian Rüdiger, Nicole Bobrowski, Alexandra Gutmann and Thorsten Hoffmann acknowledge support by the research center 'Volcanoes and Atmosphere in Magmatic, Open Systems' (VAMOS), University of Mainz, Germany. Julian Rüdiger is thankful for funding from Max Planck Graduate School at the MPIC (MPGS), Mainz, the German Academic Exchange Service

(DAAD) and the support by OVSICORI (Costa Rica) and INETER (Nicaragua). The authors also thank Ulrich Platt for his support on the theoretical assessment of multicopter sampling.

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
