# Peer review of "Implementation of electrochemical, optical and denuder-based sensors and sampling techniques on UAV for volcanic gas measurements: examples from Masaya, Turrialba and Stromboli Volcanoes"

_Atmospheric Measurement Techniques, 2017_

## Author Comment (AC1) · 15 Dec 2017

In order to reference another Sunkist application for volcanic gas measurements and the related data evaluation with the response time correction by Arellano et al. (2017) we want to revise section 3.5.1 as follows and add the reference:

Page 11 line 4: "To adapt the response times of the two sensors, the $CO_2$ signal was smoothed through convolution with a first order transfer function."

[Figure]

Revision: "To adjust the response times of the two sensors, the CO2 signal was smoothed through convolution with a first order transfer function adapted after Arellano et al. (2017), who also applied the Sunkist instrument in gas measurements in Papua New Guinea in 2016."

Figure text Fig. 5 "Figure 5: (a) Example of time series for mixing ratios of SO2 and CO2 (original data in red, resampled data in black), showing discrete gas masses at Stromboli volcano (1st flight on 05th April 2016), (b) Correlation plot for the determination of the relative time response factor for the CO2 gas sensor with a maximum at a relative time response factor of 1.7, (c) CO2 over SO2 mixing ratios, showing the outcome of the resampling of the fast CO2 with a relative time response factor of 1.7 (lower plot), linear regression results CO2/SO2 ratios of 64 ± 16 the first peak and 42 ± 4 for the second."

Revision: "Figure 5: (a) Example of time series for mixing ratios of SO2 and CO2 (original data in red, resampled data in black), showing discrete gas masses at Stromboli volcano (1st flight on 05th April 2016), (b) Correlation plot for the determination of the relative time response factor for the CO2 gas sensor with a maximum at a relative time response factor of 1.7 (adapted after Arellano et al. (2017)), (c) CO2 over SO2 mixing ratios, showing the outcome of the resampling of the fast CO2 with a relative time response factor of 1.7 (lower plot), linear regression results CO2/SO2 ratios of 64 ± 16 the first peak and 42 ± 4 for the second."

Reference:

Arellano, S., Galle B., Mulina K., Wallius, J., McCormick, B., Salem, L., D'aleo, R., Itikarai I., Tirpitz, L., Bobrowski, N., and Aiuppa, A.: Recent observations of carbon and sulfur gas emissions from Tavurvur, Bagana and Ulawun (Papua New Guinea) with a combination of ground and air-borne direct and remote sensing techniques, Abstract EGU2017-13644 presented at 2017 General Assembly, EGU, Vienna, Austria, 23-28 April.

---

## Referee Comment (RC1) · C. Kern (Referee) · 21 Dec 2017

**General comments**

This manuscript describes volcanic gas measurements performed with various sensors mounted on a rotary wing unmanned aerial system (UAS). A wide range of different sensor types were applied – in-situ gas concentrations were measured using a combination of electrochemical and optical sensors. For the first time, denuder-based

sampling was conducted from a UAS. Also, a differential optical absorption spectrometer (DOAS) was flown underneath the volcanic gas plume to derive sulfur dioxide (SO2) emission rates. The authors describe how this suite of sensors was applied at Masaya, Turrialba and Stromboli Volcanoes to help constrain the plume gas composition and gas flux, and gain insights into halogen chemistry in volcanic plumes.

The strength of this article clearly lies in the breadth of UAS applications that are described. The combination of various sensor technologies targeting different observable parameters is intriguing and will be helpful for volcanic gas geochemists in framing future experiments. The various sensors are described in sufficient detail as to allow reproduction of this or development of similar sensor payloads targeting specific research questions.

The measurement results obtained during deployment at the authors' three study sites are also quite interesting, though they are somewhat difficult to interpret due to the limited amount of available UAS data and very limited supplemental information from other sources at each study site. For example, it remains unclear why C/S ratios measured by UAS at Stromboli are systematically higher than those measured by ground-based MultiGAS during the study period. Similarly, the three BrX/SO2 ratios measured at different distances from Stromboli's active vents do not follow a clear trend and are hard to interpret by themselves. Perhaps a bit more effort could be made to put these measurement results into context and/or offer possible explanations for the observations.

The manuscript is generally well written and the technical information contained within will be useful to the volcanology community. The content is within the scope of Atmospheric Measurement Techniques, and I recommend the article be published after the specific issues and minor corrections listed below are taken into consideration.

.

Specific issues

[Figure]

Title – Currently, the main focus of the manuscript appears to be the implementation of various sensors for characterizing volcanic degassing on a UAS platform. This is also in line with the scope of AMT. Unless the results section is significantly expanded, you might consider changing the title to something along the lines of "Implementation of electro-chemical, optical and denuder-based sensors and sampling techniques on UAS for volcanic gas measurements: examples from Masaya, Turrialba and Stromboli Volcanoes". This seems to capture the manuscript's focus a bit better than the current title.

P1L32 – 'spatial and temporal proximity to explosions'? Is the spatial and temporal evolution of the C/S ratio actually discussed in the manuscript? It seems like this is a bit of a reach. Perhaps better to say that changes in the C/S ratio were observed that may have been associated with explosive activity at Stromboli?

P1L31ff – Why are only the results from Stromboli mentioned explicitly in the abstract? Perhaps the most important results for each study site could be mentioned?

P2L13 – 'It has been shown. . .' This is much too general of a statement. There are accounts of increased C/S prior to eruptions. However, the opposite has also been observed (e.g. at Poas, see your reference de Moor et al 2016b). Please clarify.

P2L27 – 'It was observed. . .' Again, I feel like this statement is too general. I think that recent measurements at Cotopaxi seemed to show an increase in BrO/SO2 during a period of continuous activity? Is this true? Dr. Bobrowski would know more of the details. . . As you mention in the next sentence, BrO is a secondary compound formed in volcanic plumes from reaction of HBr with other species. Therefore, the link between measured BrO/SO2 ratios and volcanic activity will typically be quite complex and depend on a large number of environmental conditions.

P5L5 – Clearly, Masaya is a large contributor to total arc emissions, but recently I believe that Turrialba has had similar emission rates. Dr. de Moor would know this better than I do, but characterizing Masaya as the 'single largest contributor' may no

longer be quite accurate. See de Moor et al 2017:

de Moor, J. M., Kern, C., Avard, G., Muller, C., Aiuppa, A., Saballos, A., . . . Fischer, T. P. (2017). A new sulfur and carbon degassing inventory for the Southern Central American Volcanic Arc : The importance of accurate time-series datasets and possible tectonic processes responsible for temporal variations in arc-scale volatile emissions. Geochemistry Geophysics Geosystems, 18, 1–32. https://doi.org/doi: 10.1002/2017GC007141

P5L23 – Does ash deposition really make maintenance risky or impossible? Please explain why. Obviously it makes frequent, tedious maintenance necessary. And if stations are very close to the summit, then ballistics pose a real threat that would make maintenance risky or impossible. But ash?

P7L15 – Is the light source really a 'small light bulb'? I always thought it was a diode. The datasheet says 'IR lamp' which might really be a light bulb but I'm not sure. . . Thanks for clarifying.

Table 1 – I have a few questions on information in this table: (1) the specifications on both instruments seem to require non-condensing plumes, yet the manuscript later describes problems with telemetry associated with condensed plumes. Could you comment further on the issue of condensation? How would it affect the measurements? Would you be able to determine and filter out poor quality data collected in condensed areas of the plume? Or how to deal with this? (2) I guess the 1/T temperature dependence of the $CO_2$ sensor is simply from the ideal gas law? T would then be the air temperature, correct? (3) Later on in the manuscript, you explain that the pressure dependent diffusivity of the $SO_2$ sensor membrane makes the readout insensitive to pressure, yet a (small) correction is listed here. This is probably a second-order effect, but it's probably worth pointing out for consistency. (4) I assume that 'resolution' is the precision of the sensor? If not, could you give the precision? Also, what is the assumed integration time for the values given? I assume you could improve precision

by increasing the integration time, correct?

P10L3 – Can you be a bit more specific with regards to which species can be detected with your denuder system? You mention reactive bromine (BrX). Am I correct in assuming that gaseous HBr cannot be detected? What about the other gaseous species involved in the 'bromine explosion' mechanism, i.e. Br2, Br, HOBr? They can all be detected? And what about bromine taken up onto aerosols? I guess it would be invisible to the instrument?

P11L2 and Author Comment from 15 Dec 2017 – There is significant literature on the issue of comparing data from sensors with different response times. For example, it would be good to cite one or both of these studies:

Roberts, T. J., Saffell, J. R., Oppenheimer, C., & Lurton, T. (2014). Electrochemical sensors applied to pollution monitoring: Measurement error and gas ratio bias — A volcano plume case study. Journal of Volcanology and Geothermal Research, 281, 85–96. https://doi.org/10.1016/j.jvolgeores.2014.02.023

Roberts, T. J., Braban, C. F., Oppenheimer, C., Martin, R. S., Freshwater, R. A., Dawson, D. H., . . . Jones, R. L. (2012). Electrochemical sensing of volcanic gases. Chemical Geology, 332–333, 74–91. https://doi.org/10.1016/j.chemgeo.2012.08.027

In the author comment from 15 Dec 2017, an EGU presentation is cited in this context, but I was not able to find the presentation online. Only the abstract is available, and this makes no mention of a method used to correct for different sensor response times. Also, please clarify how exactly the 'response time factor' is defined.

P11L22 – How long was the denuder sampling period? I.e. how long did the instrument need to hover in the plume to collect a good sample?

P13L14 – This is where condensed water is mentioned, despite the fact that the sensors are specified to require non-condensing conditions. Please explain the caveats with these measurements if possible.

P14L8 – At Stromboli, explosions may be associated with $CO_2$-rich gas slugs rising through the conduit and venting into the atmosphere. However, in this model, all the $CO_2$ and $SO_2$ is emitted from the vent itself. Once in the atmosphere, the gas is diluted of course, but as far as I can tell, the ratio of volcano $CO_2$ to $SO_2$ should remain constant over time and space. It is not at all clear to me why the C/S ratio would be different once the plume becomes more dilute. Please explain the mechanism that you are suggesting may change that ratio as the plume moves in space and time.

P14L14 – You state that the MultiGAS measurements broadly agree with the UAS measurements, but fail to mention that there appears to be quite a large systematic difference between the average values obtained by the two instruments. According to Figure 7, the MultiGAS seems to measure C/S of no more than 15, with an average of about 7, whereas the UAS instrument measured between about 10 and 65, with an average of around 30. This is a significant difference and should be addressed in the text. Simply stating that the measurements were not taken at the identical time and place is a little weak in terms of an explanation, especially given my previous comment.

P14L17 – I don't understand why high C/S ratios should be left aside. You do have at least some observations of ongoing eruptive activity during the time that you were there for the UAS measurements, and clearly the datasets overlap in time so in first order approximation, you would think that the same activity was sampled by both instruments. Can you please clarify?

Table 2 – I assume that the 'lower $SO_2$ limit' refers to a limit below which the data was not used for deriving C/S ratios. Can you please explain how this limit was chosen and why it varied for different datasets?

Figure 6 – Either I'm not understanding or something appears to be amiss with this figure and/or the caption. The bottom two plots are labeled the same. I assume that the bottom plot should actually be the MG $SO_2$ mixing ratio, correct? And in the caption, I assume that you mean that the SK $CO_2$ raw data is shown in grey and the resampled

CO2 data is shown in black, correct?

P17L10 – Here you point out that BrX/SO2 appears to vary with CO2/SO2, though no trend can be derived from the three obtained data points. What does this observation really mean? If BrX/SO2 was in some way proportional or anti-proportional to CO2/SO2, then one might attribute the change to various gas compositions being emitted from the volcano at different times. However, a varying dependency seems to negate this explanation as being primarily responsible. So what could possible cause this? Or is this a sign that something is wrong with the derived CO2/SO2? (also see previous comments on comparison to MultiGAS).

Figure 7 – What criteria were used to select valid MultiGAS data? You mention the different SO2 lower limits for the SK, but what about the MultiGAS? Also, as mentioned before, I think the systematic difference shown here is a bit alarming and needs some careful thought and discussion.

P20L25 – What do you mean by 'previously unstudied plume regions'? Areas very close to the vent? What do you think are the limitations on this, e.g. with regards to heat exposure, ash concentration etc.?

P20L25 – I may be wrong, but I think that UAS operations with pre-programmed flight paths have already been done, see e.g.

Mori, T., Hashimoto, T., Terada, A., Yoshimoto, M., Kazahaya, R., Shinohara, H., & Tanaka, R. (2016). Volcanic plume measurements using a UAV for the 2014 Mt. Ontake eruption the Phreatic Eruption of Mt. Ontake Volcano in 2014 5. Volcanology. Earth, Planets and Space, 68(1). https://doi.org/10.1186/s40623-016-0418-0

.

Minor corrections

The manuscript would benefit from careful proof-reading. A significant number of minor corrections would improve the legibility of the text. Listed below are some of the more

important corrections needed for clarity, but there are likely several others.

P1L18 – . . . (e.g. carbon dioxide) TO THE ATMOSPHERE.

P1L19 – Consider rewording this sentence to something like: The relative abundance of carbon and sulfur in volcanic gas as well as the total sulfur dioxide emission rate from a volcanic vent are established parameters in current volcano monitoring strategies, and they oftentimes allow insights into subsurface processes. On the other hand, chemical reactions involving halogens are thought to have local to regional impact on the atmospheric chemistry around passively degassing volcanoes.

P1L21 – Recommend removing 'on board'

P1L22 – Recommend removing 'with such new measurement strategy'

P1L23 – Consider appending the altitudes to the individual volcanoes, e.g. Turrialba Volcano (3,300 m), Stromboli Volcano (930 m) . . .

P1L27 – Remove ',' after including

P2L6 – Consider mentioning v. Glasow et al 2009 for a more complete treatise of plume chemistry? von Glasow, R., Bobrowski, N., & Kern, C. (2009). The effects of volcanic eruptions on atmospheric chemistry. Chemical Geology, 263(1–4), 131–142. https://doi.org/10.1016/j.chemgeo.2008.08.020

P2L11 – There are a few other recent articles that could be mentioned in this context:

Mason, E., Edmonds, M., & Turchyn, A. V. (2017). Remobilization of Crustal Carbon May Dominate Volcanic Arc Emissions. Science, 357, 290–294. https://doi.org/10.1126/science.aan5049

de Moor, J. M., Kern, C., Avard, G., Muller, C., Aiuppa, A., Saballos, A., . . . Fischer, T. P. (2017). A new sulfur and carbon degassing inventory for the Southern Central American Volcanic Arc : The importance of accurate time-series datasets and possible tectonic processes responsible for temporal variations in arc-scale volatile

emissions. Geochemistry Geophysics Geosystems, 18, 1–32. https://doi.org/doi:
10.1002/2017GC007141

P2L12 – the observation of gas composition changes HAS BECOME an important tool

P2L18 - . . . characterization of volcanic ACTIVITY IS GAS EMISSION RATE. Particularly, the determination of SO2 FLUX has become. . .

P2L21 - . . . manned AIRCRAFT, . . .

P2L22 - . . .poorly accessible TERRAIN.

P3L13 – crater rim MAY BE ASSOCIATED WITH a considerable . . .

P3L14 – Perhaps be more general and say that gas monitoring stations are deployed in close proximity to active volcanic vents (rather than 'at the crater rim')

P3L25 – please specify that 'DRONE-BASED sampling' has not yet been reported.

P3L27 – Change 'systems' to 'system'.

P4L7 – 'horseshoe-shaped area THAT IS NOT SAFELY ACCESSIBLE ON FOOT'

P4L8 – 'well accessible, and numerous monitoring stations have been installed here for continuous observation of the ongoing volcanic activity'

P4:14 – The UAV was mostly launched at the northern shelter

Figure 1 caption – Overview OF the sampling. . .

P5L9 – the 'DCO-DECADE' and NOVAC acronyms should probably be explained and perhaps a reference can be added where more information can be found on DCO-DECADE?

P6L9 - . . .using PROPELLERS with a diameter. . .

P6L17 . . . areas of dense plume IN WHICH to hover the system. . .

P7L1 – within a radius of a few meters AROUND THE INLET.

P7L2 – please clarify what you mean by 'which represents homogeneous conditions for a widely spread out plume'. I did not understand this phrase.

Figure 3 caption – (c) interior view OF the . . .

P9L6 . . . foam case AND HAS a total weight of 500 g.

P9L8 – Gas was pumped through the sensors in series.

P9L20 - . . . to ensure that weight requirements WERE MET.

P9L23 – change 'mixing ration' to 'mixing ratio'.

P10L21 – Consider replacing 'Evaluation' with 'Validation'

P13L7 - . . .above ground level OF 1080 m was recorded

P14L4 – These flights covered distances OF between 11 and 419 m from the vent

P14L5 – Consider changing 'gas masses' to 'gas clouds' or similar to avoid confusion with a measure of weight.

P14L8 – change 'explosion' to 'explosions'

P14L13 – Recommend removing 'has'

P14L15 – Remove 'an' before ordinary

P14L16 - . . . both instruments DID not MEASURE SIMULTANEOUSLY or. . .

P17L4 - . . . bromine SPECIATION in volcanic plumes has been THE subject of. . .

P17L8 - . . . The DATA PRESENTED HERE for the first minute after emission HIGH-LIGHT the potential. . .

P17L9 - . . . thus OBTAIN a better understanding. . .

Figure 7 caption – Perhaps include the volcano name (Stromboli) in the caption to clarify the measurement location.

P19L1 – . . . can significantly change THE PLUME'S travel direction. . .

P20L16 – Consider changing 'high-class' to 'more sophisticated'

P20L24 – . . . gain INSIGHTS into . . .

P20L25 - . . . this method COULD yield data from. . .

Supplementary Material – is there a PDF document missing here? There is mention of a PDF containing the wiring diagram of the Black Box unit, but I don't see that here. Please double check on this. Thank you!

––––––––––––––––––––––––––––––––

---

## Referee Comment (RC2) · J. A. Diaz (Referee) · 21 Dec 2017

The paper presents a very good description of the possibilities to use new UAV platforms and developed prototype instrumentation to acquire gas data that otherwise will be very difficult to obtain without the risk of human lives. It describes 3 field deployment sites with respective results demonstration the versatility of the combined capability. The paper presents what I believe is part of the beginning of a new standard of

volcanic gas emission monitoring, but is yet too soon be determine what are the right sensors, technologies and procedures to acquire the insitu data to provide meaningful information on the conditions of an active volcano. There will be a need of multiple researchers,sensors evaluations, platforms, payloads, instrument demonstration, workshops and publications to determine the best method to choose as a a new standard, so, I expect this will be a trend topic for many researchers to explore the capabilities of UAV base volcanic plume measurements with different payloads and instruments. So this makes the proposed paper pioneer and important for this field togetehr with other scientist already doing similar measurements. I do recommend the article for publication with the small corrections that the other referee is suggesting, which in most cases I concur.

---

## Short Comment (SC1) · 9 Jan 2018

R. Campion

robin@geofisica.unam

I would just like to point out that the first application of a UAV to the study of the chemistry of volcanic plumes dates back to the late 70s, and is published in this reference : Faivre-Pierret, R., Martin, D. & Sabroux, J.C. Bull Volcanol (1980) 43: 473. https://doi.org/10.1007/BF02597686. At that time, the Bulletin of Volcanology was still multinlingual and this paper, written in french with an english abstract, has gone be-

low the radar of the later paspers on volcanic plumes studied with UAVs. Summarized briefly, the authors mounted, on a fixed-wing UAV, a sampling unit composed of a pump and impregnated filters to measure the concentration of SO2, HCl and HF in the plume of Mt Etna. I believe that the authors could cite this reference, as to give credit to this pioneering work.

---

## Author Comment (AC2) · 20 Feb 2018

**Authors' response letter – AMT-2017-355**

The authors like to thank both referees for reviewing the discussion paper, and especially acknowledge the detailed comments and suggestions by Christoph Kern, which helped to significantly improve the manuscript.

**Answer to comments from Referee 1 (Christoph Kern):**

*General Comment: The measurement results obtained during deployment at the authors' three study sites are also quite interesting, though they are somewhat difficult to interpret due to the limited amount of available UAS data and very limited supplemental information from other sources at each study site. For example, it remains unclear why C/S ratios measured by UAS at Stromboli are systematically higher than those measured by ground-based MultiGAS during the study period. Similarly, the three BrX/SO2 ratios measured at different distances from Stromboli's active vents do not follow a clear trend and are hard to interpret by themselves. Perhaps a bit more effort could be made to put these measurement results into context and/or offer possible explanations for the observations.*

Reply: The referee is right, indicating that the part of measurement data interpretation in a volcanological context could be improved. Although the availability of data is concise and the focus of the manuscript is on the measurement techniques, we tried to improve the corresponding section, as much as we felt comfortable with and avoiding speculations. Detailed revision on the matter of MultiGAS and BrX/SO2 data interpretation can be found later in this response letter.

*Comment: Title – Currently, the main focus of the manuscript appears to be the implementation of various sensors for characterizing volcanic degassing on a UAS platform. This is also in line with the scope of AMT. Unless the results section is significantly expanded, you might consider changing the title to something along the lines of "Implementation of electro-chemical, optical and denuder-based sensors and sampling techniques on UAS for volcanic gas measurements: examples from Masaya, Turrialba and Stromboli Volcanoes". This seems to capture the manuscript's focus a bit better than the current title.*

Reply: We thank the referee for suggesting a new title and we followed that suggestion, changing the title to:

**"Implementation of electrochemical, optical and denuder-based sensors and sampling techniques on UAV for volcanic gas measurements: examples from Masaya, Turrialba and Stromboli Volcanoes"**

*P1L32 – 'spatial and temporal proximity to explosions'? Is the spatial and temporal evolution of the C/S ratio actually discussed in the manuscript? It seems like this is a bit of a reach. Perhaps better to say that changes in the C/S ratio were observed that may have been associated with explosive activity at Stromboli?*

Reply: The evolution of the C/S ratio associated with explosive activity is discussed rather briefly (P14L8), by stating that at Stromboli other studies found that C/S ratio changes with explosive activity (e.g. La Spina et al. 2013). We observed high C/S ratio values while flying directly above the crater and occasionally withdrawing the UAS just prior to explosions. Due to the lack of complete temporal records of the explosive activity, we can only relate the C/S ratios to photograph recordings of the explosions. With that, we were able to observe elevated C/S ratios in proximity to the explosions.

**"At Stromboli volcano, elevated CO₂/SO₂ ratios have been observed in spatial and temporal proximity to explosions by airborne in-situ measurements."**

*P1L31ff – Why are only the results from Stromboli mentioned explicitly in the abstract? Perhaps the most important results for each study site could be mentioned?*

Reply: We added the most important results for each study site as the referee suggested.

**"The new instrumental set-ups were compared with established instruments during ground-based measurements at Masaya volcano, which resulted in CO₂/SO₂ ratios of 3.6 ± 0.4. For total SO₂ flux estimations a small differential optical absorption spectroscopy (DOAS) system measured SO₂ column amounts on transversal flights below the plume at Turrialba Volcano, giving 1776 ± 1108 T/d and 1616 ± 1007 T/d of SO₂ during two traverses. At Stromboli volcano, elevated CO₂/SO₂ ratios have been observed in spatial and temporal proximity to explosions by airborne in-situ measurements. Reactive bromine to sulfur ratios of 0.19 x 10⁻⁴ to 9.8 x 10⁻⁴ were measured in-situ in the plume of Stromboli volcano downwind of the vent."**

*P2L13 – 'It has been shown. . .' This is much too general of a statement. There are accounts of increased C/S prior to eruptions. However, the opposite has also been observed (e.g. at Poas, see your reference de Moor et al 2016b). Please clarify.*

Reply: We edited that sentence for clarification

**"For instance, the CO2/SO2 emission ratio strongly varies with volcanic activity, which is associated to magma rising up a conduit. The solubility of magmatic gases is pressure dependent and different gases are released from the magma at different depths during the magma ascent, that is accompanied by pressure decrease. Gas ratio changes have been observed within the timescale of hours to weeks prior to eruptions (e.g. Giggenbach, 1975; Aiuppa et al., 2007; de Moor et al., 2016a, de Moor et al., 2016b) and their magnitude, direction, and pace are highly variable throughout different volcanic systems and state of activity."**

*P2L27 – 'It was observed. . .' Again, I feel like this statement is too general. I think that recent measurements at Cotopaxi seemed to show an increase in BrO/SO2 during a period of continuous activity? Is this true? Dr. Bobrowski would know more of the details. . . As you mention in the next sentence, BrO is a secondary compound formed in volcanic plumes from reaction of HBr with other species. Therefore, the link between measured BrO/SO2 ratios and volcanic activity will typically be quite complex and depend on a large number of environmental conditions.*

Reply: We agree that this statement might be too general. We edited the paragraph on BrO/SO2 ratios to clarify that matter. Regarding BrO/SO2 emission at Cotopaxi, Dinger et al. (2017) observed higher values of BrO/SO2 only at a declining phase of activity, while prior and during the climax of activity the BrO/SO2 was lower.

**"Although several studies observed decreases in the BrO/SO₂ ratio in advance to eruptive phases (Lübcke et al., 2014) and lower ratios during periods of continuous activity (Bobrowski and Giuffrida, 2012), it is not yet clear whether magma-gas partitioning of bromine occurs prior or after sulfur during the pressure drop associated with magma ascents {Dinger 2017 #394}. Furthermore, BrO is not a directly emitted species rather than the product of complex heterogeneous chemistry in the volcanic plume**

involving reactions with magmatic gases with entrained air (e.g. Gerlach, 2004, Bobrowski et al., 2007). The variation of BrO by plume age and a transversal distribution in the plume for this species was observed by differential optical absorption spectroscopy (DOAS) measurements (Bobrowski et al., 2007). Additionally, other reactive halogen species with oxidation states ≠ -1 (e.g. $Br_2$, $Cl_2$, BrCl and others) have been measured in-situ in the plume of Mt. Etna, Italy (Rüdiger et al., 2017) and Mt Nyamuragira {Bobrowski 2017 #395}."

*P5L5 – Clearly, Masaya is a large contributor to total arc emissions, but recently I believe that Turrialba has had similar emission rates. Dr. de Moor would know this better than I do, but characterizing Masaya as the 'single largest contributor' may no longer be quite accurate. See de Moor et al 2017:*
*de Moor, J. M., Kern, C., Avard, G., Muller, C., Aiuppa, A., Saballos, A., . . . Fischer, T. P. (2017). A new sulfur and carbon degassing inventory for the Southern Central American Volcanic Arc : The importance of accurate time-series datasets and possible tectonic processes responsible for temporal variations in arc-scale volatile emissions. Geochemistry Geophysics Geosystems, 18, 1–32. https://doi.org/doi: 10.1002/2017GC007141*

Reply: The Referee is correct. Latest emission inventories show that Turrialba volcano emissions overtook those of Masaya volcano in 2015-2016. Furthermore, more recent but unpublished measurements showed similar emission strengths for both volcanoes. The paragraphs dealing with this case were edited.

"Masaya persistently emits voluminous quantities of $SO_2$, with fluxes typically ranging from 500 T/d to 2500 T/d (e.g. (Mather et al., 2006; de Moor et al., 2013; Carn et al., 2017), making this volcano currently to one of the largest contributor to volcanic gas emissions in the Central American Volcanic Arc (de Moor et al. 2017). […] In the 2000s, an almost 150 years long period of quiescence has ended and since 2010 several vent opening phreatic eruptions henceforth occurred marking an ongoing but erratic phase of unrest (Martini et al., 2010), characterized by variable ash and gas emission intensities (up to 5000 tons/day of $SO_2$ (de Moor et al., 2016a)), making Turrialba volcano to the second substantial emitter in the arc, besides Masaya volcano (de Moor et al. 2017)."

*P5L23 – Does ash deposition really make maintenance risky or impossible? Please explain why. Obviously it makes frequent, tedious maintenance necessary. And if stations are very close to the summit, then ballistics pose a real threat that would make maintenance risky or impossible. But ash?*

Reply: We added the aspect of ballistic impacts posing threats to maintenance personnel. While ash emissions itself might not pose an imminent threat, they make maintenance necessary at higher frequencies, which leads to longer exposition periods at the crater for the researcher.

"However, stations located near the active vent suffer from ash deposition and ballistic impacts during more frequent episodes, making maintenance demanding and risky or impossible."

*P7L15 – Is the light source really a 'small light bulb'? I always thought it was a diode. The datasheet says 'IR lamp' which might really be a light bulb but I'm not sure. . . Thanks for clarifying.*

Reply: Actually, we assumed the light source to be a light bulb due to the following observations: 1.) Every time the sensor performs a measurement, one can actually see the light flashing through the diffusion membrane with the typical warm yellowish color of a thermal emitter (unfortunately a spectral analysis has not been performed). 2.) Switching the light source on and off occurs smoothly 3.) The whole sensor is very cost-effective (<100€). Even under mass-production this is unlikely to achieve with LEDs, emitting at appropriate wavelengths. Anyway, we will express it more carefully:

**"[…], where it is exposed to the radiation of a small infra-red light source"**

*Table 1 – I have a few questions on information in this table: (1) the specifications on both instruments seem to require non-condensing plumes, yet the manuscript later describes problems with telemetry associated with condensed plumes. Could you comment further on the issue of condensation? How would it affect the measurements? Would you be able to determine and filter out poor quality data collected in condensed areas of the plume? Or how to deal with this? (2) I guess the 1/T temperature dependence of the CO2 sensor is simply from the ideal gas law? T would then be the air temperature, correct? (3) Later on in the manuscript, you explain that the pressure dependent diffusivity of the SO2 sensor membrane makes the readout insensitive to pressure, yet a (small) correction is listed here. This is probably a second-order effect, but it's probably worth pointing out for consistency. (4) I assume that 'resolution' is the precision of the sensor? If not, could you give the precision? Also, what is the assumed integration time for the values given? I assume you could improve precision by increasing the integration time, correct?*

Reply: First of all, these specifications are from the manufacturers data sheets and not established by us. Also the typical applications for most of the parts are certainly not volcanic gas measurements and we like others probably expand the recommended boundaries of the parts specifications. We tried to answer the specific questions as follows:

(1) Condensed water inside the sensors would definitely affect the measurements. In the CO2 sensor, reflectivity of the optical cell would change and cause apparent absorption, for the SO2 sensor water on the electrode would change reactivity. To avoid this, we operated the instrument with a 0.45 µm pore filter at the gas inlet, retarding any particulate matter (including condensed water droplets) that might interfere with the measurement. Further, due heat dissipation of the electronics/electromechanics and the thermally isolating styrofoam housing of the Sunkist, the sensors are typically held at temperatures several degree above ambient conditions, such that condensation inside the instrument is unlikely when measuring in well diluted plumes (meaning that the sampled gases are close to ambient temperature). Therefore, the sensors inside the housing can be said to be operated at non-condensing conditions. If there is risk of condensation due to very extreme conditions (e. g. sampling hot plumes close to the vent), affected data can be identified by a) looking at the data of the humidity sensor inside the CO2 sensor housing or b) changes in the response behavior of the CO2 sensor (gas absorption signals rise and decline on sub second time scales, whereas condensation/drying of the sensor occurs much slower). Such slow responses can easily be recognized when comparing the CO2 VMR to the SO2 signal. In the case of SO2, we used the same sensor (CiTiceL) as other instrument manufacturer (e.g. INGV) and therefore applied a similar filter system in our sampling line.

(2) Correct. Also, this dependency was included in the calibration function used for the CO2 sensor.

(3) This pressure dependence again origins from the data sheet and is listed to give the complete specifications. With 0.0015% /kPa dependency a change of 1 bar would lead to 0.15% signal variation, which is negligible in applications at atmospheric pressure. Potentially important in industrial applications at unusual pressures.

(4) Indeed, "Resolution" is misleading. The given values are the 1-sigma instrument noise at a temporal resolution of 0.5 seconds, which is a measure for the short-term precision (neglecting drifting of the instrument). This precision can be improved on cost of temporal resolution by averaging. Long-term drifts are considered in the "Accuracy", also given in the table. Further, we noticed a mistake in the table: the 0.5 ppm given by the supplier seem to apply for the low sensitivity version of the sensor. Our measured noise at the high sensitivity version (sensitivity increased by a factor of 10) is approximately 0.05 ppm. We corrected this in the table.

"

| | | |
|---|---|---|
| **Accuracy** | ± 30 ppm ± 5 % signal ($CO_2$),
 ± 1 ppm ± 1 % signal ($SO_2$) | ± 1 ppm ± 1 % signal |
| **Instrument noise (1σ)** | 5 ppm ($CO_2$), 0.05 ppm ($SO_2$) | 0.05 ppm |

"

*P10L3 – Can you be a bit more specific with regards to which species can be detected with your denuder system? You mention reactive bromine (BrX). Am I correct in assuming that gaseous HBr cannot be detected? What about the other gaseous species involved in the 'bromine explosion' mechanism, i.e. Br2, Br, HOBr? They can all be detected? And what about bromine taken up onto aerosols? I guess it would be invisible to the instrument?*

Reply: We applied two coatings, of which one is sensitive to HBr. TMB on the other hand is selectively sensitive towards gaseous molecular bromine species in which the bromine atom is in oxidation state +1 or 0 (e.g. $Br_2$ or BrCl). Particulate bromine would pass the denuder "undetected", since the diffusion coefficient of particles is rather high. However, since $Br_2$ and BrCl is rather insoluble in aqueous phase the assumption is that it's primarily found in the gas phase.

**"Gas diffusion denuder sampling, which enriches gaseous compounds while being insensitive to the particle phase, was applied by using two types of coating materials as derivatization agent for the gas diffusion sampling. Total gaseous reactive molecular bromine species, BrX ($Br_2$, BrCl, (H)OBr), were determined by denuders coated with 15 µmol of 1,3,5-trimethoxybenzene (TMB) - which reacts to 1-bromo-2,4,6-trimethoxybenzene - and subsequent gas chromatography-mass spectrometry (GC-MS) analysis (Rüdiger et al., 2017)."**

*P11L2 and Author Comment from 15 Dec 2017 – There is significant literature on the issue of comparing data from sensors with different response times. For example, it would be good to cite one or both of these studies:*

*Roberts, T. J., Saffell, J. R., Oppenheimer, C., & Lurton, T. (2014). Electrochemical sensors applied to pollution monitoring: Measurement error and gas ratio bias at volcano plume case study. Journal of Volcanology and Geothermal Research, 281, 85–96. https://doi.org/10.1016/j.jvolgeores.2014.02.023*

*Roberts, T. J., Braban, C. F., Oppenheimer, C., Martin, R. S., Freshwater, R. A., Dawson, D. H., . . . Jones, R. L. (2012). Electrochemical sensing of volcanic gases. Chemical Geology, 332–333, 74–91. https://doi.org/10.1016/j.chemgeo.2012.08.027*

*In the author comment from 15 Dec 2017, an EGU presentation is cited in this context, but I was not able to find the presentation online. Only the abstract is available, and this makes no mention of a method used to correct for different sensor response times. Also, please clarify how exactly the 'response time factor' is defined.*

Reply: We replaced the former explanation by a more complete description, including the definition of the 'response time factor' and the first citation proposed by the referee (Roberts et al. 2014).

"In order to adjust the response times of the two sensors, a slow response signal for the CO2 sensor (CO2,sim) was simulated. This was achieved through convolution of the original signal with a typical sensor pulse response

$$f(t) = \begin{cases} 0, & t < 0 \\ \dfrac{1}{\tau}e^{-t/\tau}, & t \geq 0 \end{cases}$$

with $t$ being time and $\tau$ being the 'response time factor', which in this context can be regarded as a measure for the degree of smoothing. The approach is mathematically equivalent to an approach shown by Roberts et al. (2014). The response time factor $\tau$ was tuned, such that the correlation of $CO_{2,sim}$ and $SO_2$ signal got maximized for discrete peaks (see Fig. 5 (a)). This was already done by Arellano et al. (2017), who also applied the Sunkist instrument in gas measurements in Papua New Guinea in 2016."

Figure 5: (a) Example of time series for mixing ratios of $SO_2$ and $CO_2$ (original data in red, resampled data in black), showing discrete gas masses at Stromboli volcano (1st flight on 05th April 2016), (b) Correlation plot for the determination of the relative time response factor for the $CO_2$ gas sensor with a maximum at a relative time response factor of 1.7, (c) $CO_2$ over $SO_2$ mixing ratios, showing the outcome of the resampling of the fast $CO_2$ with a relative time response factor of 1.7 (lower plot), linear regression results $CO_2/SO_2$ ratios of 64 ± 16 the first peak and 42 ± 4 for the second.

*P11L22 – How long was the denuder sampling period? I.e. how long did the instrument need to hover in the plume to collect a good sample?*

Reply: The sampling period is shown in table 4. We sampled between 1 and 5 minutes. Depending on the mixing ratio of reactive molecular bromine species and flow rates the sampling times were sufficient to trap about 1 to 3 ng of Br2 equivalents, which lead to detectable and quantifiable signals in with the used GC-MS method including a previous pre-concentration step during the sample preparation.

*P13L14 – This is where condensed water is mentioned, despite the fact that the sensors are specified to require non-condensing conditions. Please explain the caveats with these measurements if possible.*

Reply: As mentioned earlier, we used µm pore sized filters to filter out condensed/particle phase at the Sunkist instrument. At Masaya, we also compared the $SO_2$ sensors signals for both instruments (Sunkist and Black Box) and could not find significant differences. Most of the sensors used in volcanological applications are probably not specifically designed by the manufacturer for those environments and I would assume that the specifications from the data sheet are mostly for user, who do not calibrate or use the sensors in scientific applications.

*P14L8 – At Stromboli, explosions may be associated with CO2-rich gas slugs rising through the conduit and venting into the atmosphere. However, in this model, all the CO2 and SO2 is emitted from the vent itself. Once in the atmosphere, the gas is diluted of course, but as far as I can tell, the ratio of volcano CO2 to SO2 should remain constant over time and space. It is not at all clear to me why the C/S ratio would be different once the plume becomes more dilute. Please explain the mechanism that you are suggesting may change that ratio as the plume moves in space and time.*

Reply: The referee is correct here. Dilution does not change the C/S ratio. The statement might be phrased a bit unfortunate. It is stated that we measured high C/S values close to the vent in an undiluted plume region and not that it changes by dilution.

**"[…] and therefore act as a possible explanation for the detected high $CO_2/SO_2$ ratios."**

*P14L14 – You state that the MultiGAS measurements broadly agree with the UAS measurements, but fail to mention that there appears to be quite a large systematic difference between the average values obtained by the two instruments. According to Figure 7, the MultiGAS seems to measure C/S of no more than 15, with an average of about 7, whereas the UAS instrument measured between about 10 and 65, with an average of around 30. This is a significant difference and should be addressed in the text. Simply stating that the measurements were not taken at the identical time and place is a little weak in terms of an explanation, especially given my previous comment.*

Reply: We agreed with the referee that this issue could be addressed a bit more in detail. We state that our UAS measurement agree with only some of the MultiGAS measurements. The MultiGAS instrument only measures four times a day for 30 minutes and averages over this period, which already leads to attenuation of high C/S ratios associated with CO2 rich gas bubbles, e.g. when a gas cloud with high C/S ratios passes the instrument only for a period of seconds. Another point we made is that we observed the explosion, while the UAS was flying in direct proximity to the vent, while there is no record of the explosions for the MultiGAS data, because they were not measuring the plume at that time of the day due to the discontinuous monitoring (4 times a day) and the wind direction wasn't blowing to the MultiGAS station at this moment. Both instruments only present "snapshot" data, but of a different kind.

**"The MG instrument only measures twice a day for 30 minutes and averages over this time, while with the SK instrument we identified discrete peaks of gas clouds with different $CO_2/SO_2$ ratios."**

*P14L17 – I don't understand why high C/S ratios should be left aside. You do have at least some observations of ongoing eruptive activity during the time that you were there for the UAS measurements, and clearly the datasets overlap in time so in first order approximation, you would think that the same activity was sampled by both instruments. Can you please clarify?*

Reply: Hopefully the explanation above already clarified that matter. For a comparison of MG and SK, high values should be left aside since we do not know whether the MG measured any eruptive released gases. While those gas clouds probably only make up a minor period for the MultiGAS integration time.

*Table 2 – I assume that the 'lower SO2 limit' refers to a limit below which the data was not used for deriving C/S ratios. Can you please explain how this limit was chosen and why it varied for different datasets?*

Reply: For the calculation of the C/S ratios, we used a linear regression with error propagation (York et al. 2004). The size of the error obtained by this propagation is partly dependent from the number of data points. With a large number of data points (associated with a longer exposition to the same gas cloud) lower limits could be used for the scatter plot and linear regression, since the noisier signal at lower ppm values do not contribute significantly to the error propagation. On the other hand, for gas clouds which only briefly pass the sensors (or the sensors the clouds), we chose to only use the less noisy signals, meaning a higher SO2 limit to derive the ratio, since potential outliers would affect the C/S ratio and its

error more. Lower limits are accompanied with larger errors and therefore lower confidence, which is a drawback related to short exposition times in UAS operations.

*Figure 6 – Either I'm not understanding or something appears to be amiss with this figure and/or the caption. The bottom two plots are labeled the same. I assume that the bottom plot should actually be the MG SO2 mixing ratio, correct? And in the caption, I assume that you mean that the SK CO2 raw data is shown in grey and the resampled CO2 data is shown in black, correct?*

Reply: We thank the referee for pointing out the flaws in the figure and caption.

**"Figure 6: Comparison of $SO_2$ and $CO_2$ time series of a Multi-GAS (MG) instrument and the *Sunkist* (SK) unit at the Masaya volcano crater rim (for SK $CO_2$ raw data in grey, resampled data in black), both instruments inlets were place in proximity to each other (14th July 2016); SK $CO_2/SO_2$ = 3.63+/- 0.43 background $CO_2$ = 439 ppm); MG $CO_2/SO_2$ = 2.94 +/- 0.30 (background $CO_2$ = 413 ppm); (additional scatter plots in the supplementary material)"**

*P17L10 – Here you point out that BrX/SO2 appears to vary with CO2/SO2, though no trend can be derived from the three obtained data points. What does this observation really mean? If BrX/SO2 was in some way proportional or anti-proportional to CO2/SO2, then one might attribute the change to various gas compositions being emitted from the volcano at different times. However, a varying dependency seems to negate this explanation as being primarily responsible. So what could possible cause this? Or is this a sign that something is wrong with the derived CO2/SO2? (also see previous comments on comparison to MultiGAS).*

Reply: As we stated in the text, an interpretation would be too ambitious with the few data we obtained. What we meant to show here is that we simply proofed the principle of our techniques and can use them to obtain that kind of data (BrX/SO2 vs. CO2/SO2 or time/distance). For the investigation of volcanological and atmospheric dependencies, a larger data set would be needed.

**"CHANGES"**

*Figure 7 – What criteria were used to select valid MultiGAS data? You mention the different SO2 lower limits for the SK, but what about the MultiGAS? Also, as mentioned before, I think the systematic difference shown here is a bit alarming and needs some careful thought and discussion.*

Reply: The MultiGAS data was derived by standard procedure (RatioCalc) and a lower limit of 4 ppm of SO2. The difference in this plot seems significant on the first view, for sure. But as discussed above the difference results from different measurement periods. The MultiGAS Data represents 5 days, while the UAS data only a few hours on different days and also shows C/S ratios obtained for discrete gas clouds.

**"CHANGES"**

*P20L25 – What do you mean by 'previously unstudied plume regions'? Areas very close to the vent? What do you think are the limitations on this, e.g. with regards to heat exposure, ash concentration etc.?*

Reply: With regards to spectroscopic methods, "unstudied" seems to miss the point we wanted to make. We rather meant 'previously physically inaccessible plume regions'. Heat exposure and ash concentration

for sure limit UAS operations, but UAS could for sure fly to regions, which manned aircraft aren't able to access. In general, I would think that heat is a minor problem, since hot gas is diluted rather quickly and at Masaya drone flights have been made into Santiago crater into the gas above the lava lake, few hundred meters of distance though. After some flights at Turrialba we observed ash deposition on the UAS after flying into the plume for in-situ sampling, which did not affect the drone. After 50+ in-plume flights we now consider to change the motors of the drone.

**"CHANGES"**

*P20L25 – I may be wrong, but I think that UAS operations with pre-programmed flight paths have already been done, see e.g.*
*Mori, T., Hashimoto, T., Terada, A., Yoshimoto, M., Kazahaya, R., Shinohara, H., & Tanaka, R. (2016). Volcanic plume measurements using a UAV for the 2014 Mt. Ontake eruption the Phreatic Eruption of Mt. Ontake Volcano in 2014 5. Volcanology. Earth, Planets and Space, 68(1). https://doi.org/10.1186/s40623-016-0418-0*

Reply: You are correct. We already cited that reference. The emphasis of the statement at P20L25 was on the scheduled pre-preprogrammed flights. We were imagining hangars of UAVs at a volcano from which autonomous operations take place on a regular basis, rather than field campaigns. We rephrased that sentence.

**"Technological advances promise to enable scheduled pre-programmed and autonomous UAV operations (e.g. from hangars close to volcanoes) with extended flight times for regular hazard assessments."**

*Minor corrections*

*The manuscript would benefit from careful proof-reading. A significant number of minor corrections would improve the legibility of the text. Listed below are some of the more important corrections needed for clarity, but there are likely several others.*

Reply: We thank the referee for his help improving the manuscript and we corrected the specific items as follows.

**"CHANGES"**

*P1L18 – . . . (e.g. carbon dioxide) TO THE ATMOSPHERE.*

**"(e.g. carbon dioxide) to the atmosphere."**

*P1L19 – Consider rewording this sentence to something like: The relative abundance of carbon and sulfur in volcanic gas as well as the total sulfur dioxide emission rate from a volcanic vent are established parameters in current volcano monitoring strategies, and they oftentimes allow insights into subsurface processes. On the other hand, chemical reactions involving halogens are thought to have local to regional impact on the atmospheric chemistry around passively degassing volcanoes.*

*"The relative abundance of carbon and sulfur in volcanic gas as well as the total sulfur dioxide emission rate from a volcanic vent are established parameters in current volcano monitoring strategies, and they oftentimes allow insights into subsurface processes. On the other hand, chemical reactions involving halogens are thought to have local to regional impact on the atmospheric chemistry around passively degassing volcanoes."*

*P1L21 – Recommend removing 'on board'*

**"[…] payloads for the compositional analysis […]"**

*P1L22 – Recommend removing 'with such new measurement strategy'*

**"[…] The various applications and their potential are presented and discussed on example studies at three […]"**

*P1L23 – Consider appending the altitudes to the individual volcanoes, e.g. Turrialba Volcano (3,300 m), Stromboli Volcano (930 m) . . .*

Reply: For the abstract we think it's sufficient that the flight heights are mentioned and not the specific volcano elevation, which are given in the text already.

*P1L27 – Remove ',' after including*

**"[…] including abundances […]"**

*P2L6 – Consider mentioning v. Glasow et al 2009 for a more complete treatise of plume chemistry? von Glasow, R., Bobrowski, N., & Kern, C. (2009). The effects of volcanic eruptions on atmospheric chemistry. Chemical Geology, 263(1–4), 131–142. https://doi.org/10.1016/j.chemgeo.2008.08.020*
Reply: We thank the referee for this suggestion. We added the reference later in the text

**"Furthermore, knowledge about in-plume chemical reactions can be drawn from compositional assessment of the gases, which also helps understanding their impact on atmospheric chemistry (e.g. Lee et al., 2005; von Glasow, 2009; Gliß et al., 2015)."**

*P2L11 – There are a few other recent articles that could be mentioned in this context: Mason, E., Edmonds, M., & Turchyn, A. V. (2017). Remobilization of Crustal Carbon May Dominate Volcanic Arc Emissions. Science, 357, 290–294. https://doi.org/10.1126/science.aan5049*

*de Moor, J. M., Kern, C., Avard, G., Muller, C., Aiuppa, A., Saballos, A., . . . Fischer, T. P. (2017). A new sulfur and carbon degassing inventory for the Southern Central American Volcanic Arc : The importance of accurate time-series datasets and possible tectonic processes responsible for temporal variations in arc-scale volatile emissions. Geochemistry Geophysics Geosystems, 18, 1–32. https://doi.org/doi:10.1002/2017GC007141*

**"Measuring the emitted gas composition can provide crucial information on understanding subsurface processes related to activity changes (e.g. Allard et al., 1991; Aiuppa et al., 2007; Bobrowski and Giuffrida, 2012; de Moor et al., 2016a; Liotta et al., 2017) and help to estimate fluxes of the geological carbon cycle (e.g. Burton et al., 2013; Mason et al., 2017) and tectonic processes controlling volcanic degassing (e.g. Aiuppa et al., 2017; de Moor et al. 2017)."**

*P2L12 – the observation of gas composition changes HAS BECOME an important tool*

**"[…] the observation of gas composition changes has become an important tool for detecting […]"**

*P2L18 - . . . characterization of volcanic ACTIVITY IS GAS EMISSION RATE. Particularly, the determination of SO2 FLUX has become. . .*

**"Another important parameter for the characterization of volcanic activity is gas emission rates. Particularly, the determination of $SO_2$ flux has become a standard procedure"**

*P2L21 - . . . manned AIRCRAFT, . . .*

**"or manned aircraft, but"**

*P2L22 - . . .poorly accessible TERRAIN.*

**"accessible terrain."**

*P3L13 – crater rim MAY BE ASSOCIATED WITH a considerable . . .*

**"crater rim may be associated with considerable  risk."**

*P3L14 – Perhaps be more general and say that gas monitoring stations are deployed in close proximity to active volcanic vents (rather than 'at the crater rim')*

**"gas monitoring stations are deployed and maintained in close proximity to active volcanic vents by researchers, putting themselves at risks."**

*P3L25 – please specify that 'DRONE-BASED sampling' has not yet been reported.*

Reply: This sentence was removed.

*P3L27 – Change 'systems' to 'system'.*

**"[…] sensing (electrochemical/optical sensors) systems for the determination […]"**

*P4L7 – 'horseshoe-shaped area THAT IS NOT SAFELY ACCESSIBLE ON FOOT'*

**"horseshoe-shaped area (Sciara del Fuoco) that is not safely accessible on foot."**

*P4L8 – 'well accessible, and numerous monitoring stations have been installed here
for continuous observation of the ongoing volcanic activity'*

**"The summit above the craters is well accessible, and numerous monitoring stations have been installed here for continuous observation of the ongoing volcanic activity"**

*P4:14 – The UAV was mostly launched at the northern shelter*

**"The UAV was mostly launched at the northern shelter (see Fig. 1)."**

*Figure 1 caption – Overview OF the sampling. . .*

**"Figure 1: Overview of the sampling […]"**

*P5L9 – the 'DCO-DECADE' and NOVAC acronyms should probably be explained and perhaps a reference can be added where more information can be found on DCO-DECADE?*

**"Continuous monitoring of the gas emissions is realized by a stationary Multi-GAS (MG) system (through the Deep Carbon Observatory – Deep Earth Carbon Degassing initiative (DCO-DECADE)) at the crater rim and two scanning DOAS instruments (Network for Observation of Volcanic and Atmospheric Change (NOVAC) (Galle et al., 2010)) in the downwind direction."**

*P6L9 - . . .using PROPELLERS with a diameter. . .*

**"using propellers with"**

*P6L17 . . . areas of dense plume IN WHICH to hover the system. . .*

**"[…] dense plume in which to hover […]"**

*P7L1 – within a radius of a few meters AROUND THE INLET.*

**"[…] within a radius of a few meters around the inlet […]"**

*P7L2 – please clarify what you mean by 'which represents homogeneous conditions for a widely spread out plume'. I did not understand this phrase. Figure 3 caption – (c) interior view OF the . . .*

Reply: If the plume is widely spread we assume that the UAS samples air which represents the plume and is not attracting ambient air towards its inlet.

**"[…] represents homogeneous plume gas for a widely spread out plume […]"**
**"[…] interior view of the Sunkist […]."**

*P9L6 . . . foam case AND HAS a total weight of 500 g.*

**"[…] foam case and has a total weight […]"**

*P9L8 – Gas was pumped through the sensors in series.*

**"[…] pumped through to the sensors in series […]"**

*P9L20 - . . . to ensure that weight requirements WERE MET.*

**"[…] ensure that weight requirements were met."**

*P9L23 – change 'mixing ration' to 'mixing ratio'.*

**"[…] SO2 mixing ratio […]"**

*P10L21 – Consider replacing 'Evaluation' with 'Validation'*

**"Validation of the SO2 fluxes obtained […]"**

*P13L7 - . . .above ground level OF 1080 m was recorded*

**"[…] ground level of 1080 m was […]"**

*P14L4 – These flights covered distances OF between 11 and 419 m from the vent*

**"[…] distances of between […]"**

*P14L5 – Consider changing 'gas masses' to 'gas clouds' or similar to avoid confusion with a measure of weight.*

**"[…] gas clouds […]"**

*P14L8 – change 'explosion' to 'explosions'*

**"[…] strombolian ash explosions occurred […]"**

*P14L13 – Recommend removing 'has'*

**"[…] Multi-GAS station (placed on the SE rim of the crater terrace) discontinuously measured […]"**

*P14L15 – Remove 'an' before ordinary*

**"[…] exemplary for ordinary Strombolian activity […]"**

*P14L16 - . . . both instruments DID not MEASURE SIMULTANEOUSLY or. . .*

**"It has to be taken into account that both instruments did not measure simultaneously or in proximity to each other."**

*P17L4 - . . . bromine SPECIATION in volcanic plumes has been THE subject of. . .*

**"Although the bromine speciation in volcanic […]"**

*P17L8 - . . . The DATA PRESENTED HERE for the first minute after emission HIGHLIGHT the potential. . .*

**"The DATA presented here for the first minute after emission highlight the potential […]"**

*P17L9 - . . . thus OBTAIN a better understanding. . .*

**"acquisition and thus obtain a better understanding […]"**

*Figure 7 caption – Perhaps include the volcano name (Stromboli) in the caption to clarify the measurement location.*

**[…] system at Stromboli volcano […]**

*P19L1 – . . . can significantly change THE PLUME'S travel direction. . .*

**"[…] can significantly change the plume's travel direction […]"**

*P20L16 – Consider changing 'high-class' to 'more sophisticated'*

**"[…] more sophisticated components […]"**

*P20L24 – . . . gain INSIGHTS into . . .*

**"[…] to gain insights into reactive […]"**

*P20L25 - . . . this method COULD yield data from. . .*

**"[…] this method could yield data […]"**

*Supplementary Material – is there a PDF document missing here? There is mention of a PDF containing the wiring diagram of the Black Box unit, but I don't see that here. Please double check on this. Thank you!*

Reply: We thank the referee for hinting at the missing document. It will be added to the supplementary material

**Answer to comments from Referee 2 (J. A. Diaz):**

*"… . I do recommend the article for publication with the small corrections that the other referee is suggesting, which in most cases I concur."*

Reply: We thank the J.A. Diaz for reviewing the manuscript.

**Answer to short comment by R. Campion:**

*"I would just like to point out that the first application of a UAV to the study of the chemistry of volcanic plumes dates back to the late 70s…"*

Reply: We thank R. Campion for the reference suggestion and added it to the manuscript

**"As in the last decade with the development of compact and cost effective unmanned aerial vehicles (UAV) several deployments of gas sensors and other in-situ methods (e.g. particle detection (Altstädter et al., 2015)) as well as applications of spectrometers were realized (e.g. McGonigle et al., 2008, Diaz et al., 2015, Mori et al., 2016, Villa et al., 2016 and references therein). Pioneering UAV deployments were conducted in the late 70's (Faivre-Pierret et al, 1980)."**